# HINGE POLICY OPTIMIZATION: RETHINKING POLICY IMPROVEMENT AND REINTERPRETING PPO

## ABSTRACT

Policy optimization is a fundamental principle for designing reinforcement learning algorithms, and one example is the proximal policy optimization algorithm with a clipped surrogate objective (PPO-clip), which has been popularly used in deep reinforcement learning due to its simplicity and effectiveness. Despite its superior empirical performance, PPO-clip has not been justified via theoretical proof up to date. This paper proposes to rethink policy optimization and reinterpret the theory of PPO-clip based on hinge policy optimization (HPO), called to improve policy by hinge loss in this paper. Specifically, we first identify sufficient conditions of state-wise policy improvement and then rethink policy update as solving a large-margin classification problem with hinge loss. By leveraging various types of classifiers, the proposed design opens up a whole new family of policy-based algorithms, including the PPO-clip as a special case. Based on this construct, we prove that these algorithms asymptotically attain a globally optimal policy. To our knowledge, this is the first ever that can prove global convergence to an optimal policy for a variant of PPO-clip. We corroborate the performance of a variety of HPO algorithms through experiments and an ablation study.

## 1 INTRODUCTION

Reinforcement learning (RL) has served as a powerful framework for achieving optimal sequential decision making by directly interacting with the environment and learning from the underlying random process. *Policy optimization*, as a fundamental design principle of RL algorithms, iteratively searches for an optimal policy by alternating between a *policy evaluation* step and a *policy improvement* subroutine. Most of the popular policy optimization approaches, including the policy gradient methods (Sutton et al., 1999; Mnih et al., 2016; Silver et al., 2014; Lillicrap et al., 2016), Trust Region Policy Optimization (Schulman et al., 2015), and Proximal Policy Optimization (PPO) (Schulman et al., 2017), aim to achieve policy improvement in terms of the expected total reward via gradient-based parameter updates. Despite the empirical success of the above approaches, these policy improvement schemes inherently lead to the following fundamental issues: (i) Global convergence is usually difficult to establish due to the non-convexity of the underlying objective function (e.g., total discounted reward) and the use of gradient methods, even for the tabular cases; (ii) The policy update direction for improvement is tightly coupled with the true discounted state visitation distribution, which is usually intractable to obtain and needs to be addressed via sampling in practice.

To address the above issues, we propose to rethink policy improvement in RL from the perspective of *state-wise policy improvement*, which aims for improvement directly in the policy space through a partial ordering of the policies, as adopted by policy iteration for the optimal control of Markov decision processes (MDPs). We start by identifying two useful sufficient conditions for state-wise policy improvement and thereafter pinpoint that improvement can be achieved based solely on the sign of the advantage function. Based on this insight, we propose *Hinge Policy Optimization* (HPO) by connecting state-wise policy improvement to solving a large-margin classification problem, where we regard the process of policy improvement as training a binary classifier with hinge loss via empirical risk minimization. As policy improvement of HPO is not enforced in the expected total reward but in a state-wise manner, the policy update of HPO is completely decoupled from the state visitation distribution, which is by contrast required by many existing popular policy optimization methods. By leveraging various types of classifiers, the proposed HPO framework opens up a whole new family of policy-based RL algorithms. Interestingly, the popular PPO algorithm with a surrogate

clipped objective (PPO-clip) can be shown to be a special case of HPO with a specific type of classifier. Given the policy improvement property of the proposed classification-based scheme, we are able to establish global convergence to an optimal policy for the HPO algorithms. To the best of our knowledge, our analysis provides the first global convergence guarantee for a variant of PPO-clip.

This paper is meant to provide a clear picture of the RL algorithms in the new HPO family with global convergence guarantees and thereby introduce a new perspective for reinterpreting the popular PPO-clip algorithm. The main contributions of this paper can be summarized as follows:

- We propose HPO, a new policy optimization framework where the policy update is built on state-wise policy improvement via hinge loss. The members in the HPO family share a generic loss function, and their differences lie in the choice of the margin and the classifier. We also show that the widely-used PPO-clip algorithm can be viewed as a special case in this family.

- This paper is the first ever that can prove state-wise policy improvement and global convergence for a variant of PPO-clip algorithm[1]. Specifically, we first present a variant of the PPO algorithm *with an adaptive clipped objective*, which can be viewed as an HPO algorithm with adaptive margin, called *HPO-AM*. We prove the convergence to an optimal policy for HPO-AM, and proceed to generalize the global convergence result to some other HPO-AM like algorithms equipped with other classifiers. We also empirically validate the proposed theoretical framework through experiments and thereby corroborate the performance of various HPO algorithms with different classifiers and margins (with the experimental results provided in Appendices E and F).

## 2 MAIN RESULTS

In this section, we first review the theoretical background about RL and view PPO-clip update as a classification problem. Then, we establish global convergence for the proposed HPO-AM.

### 2.1 BACKGROUND

A discounted Markov Decision Process (MDP) is defined by the tuple $(\mathcal{S}, \mathcal{A}, \mathcal{P}, r, \gamma)$, where $\mathcal{S}$ is a *finite* state space, $\mathcal{A}$ is a *finite* action space, $\mathcal{P} : \mathcal{S} \times \mathcal{A} \times \mathcal{S} \to [0, 1]$ is the state transition probability matrix, $r : \mathcal{S} \times \mathcal{A} \to \mathbb{R}$ is the reward function, and $\gamma \in (0, 1)$ is the discounted factor. A stochastic policy $\pi : S \to \Delta(\mathcal{A})$ specifies the action distribution based on the current state, where $\Delta(\mathcal{A})$ is the probability simplex over $\mathcal{A}$, i.e., $\Delta(\mathcal{A}) = \{x \in \mathbb{R}^{|\mathcal{A}|} \mid x_0 + ... + x_{|\mathcal{A}|-1} = 1, x_i \geq 0, \forall i = 0, ..., |\mathcal{A}| - 1\}$. Given any policy $\pi$, the state value function $V^\pi$, the state-action value function $Q^\pi$, and the advantage function $A^\pi$ are defined as follows.

$$V^\pi(s) := \mathop{\mathbb{E}}_{\substack{a_t \sim \pi(\cdot|s_t), \\ s_{t+1} \sim \mathcal{P}(\cdot|s_t, a_t)}} \left[ \sum_{t=0}^\infty \gamma^t r(s_t, a_t) \Big| s_0 = s \right], \tag{1}$$

$$Q^\pi(s, a) := \mathop{\mathbb{E}}_{\substack{a_t \sim \pi(\cdot|s_t), \\ s_{t+1} \sim \mathcal{P}(\cdot|s_t, a_t)}} \left[ \sum_{t=0}^\infty \gamma^t r(s_t, a_t) \Big| s_0 = s, a_0 = a \right], \tag{2}$$

$$A^\pi(s, a) := Q^\pi(s, a) - V^\pi(s). \tag{3}$$

Define $d_s^\pi(s') := (1 - \gamma) \sum_{t=0}^\infty \gamma^t \Pr(s_t = s'|s_0 = s, \pi)$ as the normalized discounted state visitation frequency, which represents the probability of visiting state $s'$ in a trajectory of $\pi$, given that $s_0 = s$. With the above definition, (Kakade & Langford, 2002) quantified the difference in performance between two policies as follows. Given policies $\pi_1$ and $\pi_2$,

$$V^{\pi_1}(s) - V^{\pi_2}(s) = \frac{1}{1 - \gamma} \sum_{s' \in \mathcal{S}} d_s^{\pi_1}(s') \sum_{a \in \mathcal{A}} \pi_1(a|s') A^{\pi_2}(s', a). \tag{4}$$

---

[1]Regarding the convergence of PPO, (Liu et al., 2019) proves global convergence in expected total reward for a neural variant of PPO with adaptive Kullback-Leibler penalty (PPO-KL). Given the salient algorithmic difference between PPO-KL and PPO-clip, to the best of our knowledge, there remains no proof of global convergence to an optimal policy for PPO-clip.

The TRPO algorithm proceeds to maximize the expected value of surrogate function of (4) over the initial state distribution under the constraint of KL divergence. State-wise policy improvement is formalized based on the following partial ordering relation.

**Definition 1** (Partial ordering over policies). *Let $\pi_1$ and $\pi_2$ be two policies. Then, $\pi_1 \geq \pi_2$, called $\pi_1$ improves upon $\pi_2$, if and only if $V^{\pi_1}(s) \geq V^{\pi_2}(s)$, $\forall s \in \mathcal{S}$. Moreover, we say $\pi_1 > \pi_2$, called $\pi_1$ strictly improves upon $\pi_2$, if and only if $\pi_1 \geq \pi_2$ and there exists at least one state $s$ such that $V^{\pi_1}(s) > V^{\pi_2}(s)$.*

**Definition 2** (An optimal policy). *A policy $\pi^*$ is said to be an optimal policy if $\pi^* \geq \pi'$, for every policy $\pi'$. Moreover, we let $V^*(s)$ denote the optimal value of each state $s$, i.e., $V^*(s) = V^{\pi^*}(s)$.*

We first introduce two sufficient conditions for state-wise policy improvement in Propositions 1-2. These two conditions have been identified by (Hu et al., 2020, Section 3.1).

**Proposition 1.** *Given policies $\pi_1$ and $\pi_2$, $\pi_1$ improves upon $\pi_2$ if the following condition holds:*

$$\sum_{a \in \mathcal{A}} \pi_1(a|s) A^{\pi_2}(s, a) \geq 0, \ \forall s \in \mathcal{S}. \tag{5}$$

**Proposition 2.** *Given policies $\pi_1$ and $\pi_2$, $\pi_1$ improves upon $\pi_2$ if the following condition holds:*

$$(\pi_1(a|s) - \pi_2(a|s)) A^{\pi_2}(s, a) \geq 0, \ \forall (s, a) \in \mathcal{S} \times \mathcal{A}. \tag{6}$$

Proposition 1 holds for the following reason: Since all $d_s^{\pi_1}(s')$ are non-negative, all values in (4) are non-negative, and hence $\pi_1$ improves upon $\pi_2$. Proposition 2 can be derived directly from Proposition 1 and the fact that $\sum_{a \in \mathcal{A}} \pi_2(a|s) A^{\pi_2}(s, a) = 0, \ \forall s \in \mathcal{S}$.

Notably, Proposition 2 offers a useful insight that state-wise policy improvement can be achieved by determining the sign of the advantage of each state-action pair (regardless of its magnitude) and adjusting the action probabilities accordingly. In this way, no additional constraints, such as the KL divergence constraint used in TRPO (Schulman et al., 2015), are needed to ensure policy improvement. This also naturally motivates the design of using the signs of the advantage function as labels in determining the direction of the policy update. More specifically, we can draw an analogy between (6) in Proposition 2 and the training of a linear classifier: (i) The state-action pair serves as the feature vector of a training sample; (ii) The sign of $A^{\pi_2}(s, a)$ plays the role of a binary label; (iii) $\pi_1(a|s) - \pi_2(a|s)$ resembles the prediction of a linear classifier. In the next section, we substantiate this insight and present the proposed HPO framework.

In the rest of this paper, our analysis relies on the following assumptions:

**Assumption 1** (Bounded reward). *To avoid trivial cases, we assume not all rewards are zero. Since both state and action spaces are finite, it holds naturally that there exists a positive constant $R = \sup_{(s,a) \in \mathcal{S} \times \mathcal{A}} |r(s, a)| > 0$.*

**Assumption 2** (Tabular policies). *Policies are parameterized by $\pi(a|s) = \theta_{s,a}$, where $\theta_s \in \Delta(\mathcal{A})$ refers to the vector of $\theta_{s,\cdot}$ for some fixed state $s$, and $\theta \in \Delta(\mathcal{A})^{|\mathcal{S}|}$, i.e., $\theta$ is subject to $\theta_{s,a} \geq 0$ and $\sum_{a \in \mathcal{A}} \theta_{s,a} = 1, \ \forall s \in \mathcal{S}, \ \forall a \in \mathcal{A}$.*

**Notations.** Throughout this paper, we let $\langle a, b \rangle$ and $a \circ b$ denote the inner product and the Hadamard product of two real vectors $a, b$, respectively.

## 2.2 HINGE POLICY OPTIMIZATION AND PPO-CLIP

In this section, we build on the insight offered by Proposition 2 and formally present the HPO framework. To better describe HPO, we use PPO-clip as an exemplary algorithm by reinterpreting the policy update of PPO-clip as a large-margin classification problem.

**PPO-clip.** Since the TRPO algorithm can be costly, e.g., requiring the computation of multiple Hessian-vector product, (Schulman et al., 2017) proposes two versions of PPO, PPO with KL penalty (PPO-KL) and PPO with clipped surrogate objective (PPO-clip). PPO-KL replaces the KL constraints of TRPO with KL penalty and a certain surrogate objective forms a lower bound on the performance of the policy, while PPO-clip further drops the KL penalty and instead directly clips the probability ratio. As a result, the proximal term becomes inherent in the clip function. PPO-KL and

TRPO have been guaranteed to monotonically improve the expected total reward (Liu et al., 2019) over a given initial state distribution. However, there remains no theoretical analysis for PPO-clip.

The original objective function of PPO-clip (Schulman et al., 2017) is to maximize the following:

$$L^{\text{clip}}(\theta) = \mathbb{E}_{s \sim d_{\mu_0}^{\pi}, a \sim \pi(\cdot|s)} \left[ \min\{\rho_{s,a}(\theta)A^{\pi}(s,a), \text{clip}(\rho_{s,a}(\theta), 1 - \epsilon, 1 + \epsilon)A^{\pi}(s,a)\} \right]$$

$$= \sum_{s \in \mathcal{S}} d_{\mu_0}^{\pi}(s) \sum_{a \in \mathcal{A}} \pi(a|s) \min\{\rho_{s,a}(\theta)A^{\pi}(s,a), \text{clip}(\rho_{s,a}(\theta), 1 - \epsilon, 1 + \epsilon)A^{\pi}(s,a)\}, \quad (7)$$

where $\rho_{s,a}(\theta)$ denotes the probability ratio $\frac{\pi_{\theta}(a|s)}{\pi(a|s)}$, $\mu_0$ the initial state distribution, $\pi(a|s)$ the old policy, $\pi_{\theta}(a|s)$ the new policy parameterized by $\theta$, $d_{\mu_0}^{\pi}(s)$ the normalized discounted state visitation frequency, and $\epsilon$ the clipping range. The function $\text{clip}(\rho_{s,a}(\theta), 1 - \epsilon, 1 + \epsilon)$ in (7) modifies the objective by removing the incentive for moving the ratio $\rho_{s,a}(\theta)$ outside of the interval $[1 - \epsilon, 1 + \epsilon]$.

**Connecting PPO-clip and Hinge Loss.** In PPO-clip, the policy stops being updated when the probability ratio is out of the clipping range. That is, the subgradient method keeps "pushing" the new policy away from the old one until it "improves" the old policy by a margin. This behavior coincides with the large-margin classification problem where the classifier intends to "push" the predicted label out of a margin. (Pi et al., 2020) has shown that maximizing each term in the empirical average objective function of PPO-clip is equivalent to minimizing a hinge loss times the magnitude of advantage as follows. Specifically, the equivalence actually refers to their gradients instead of the functions themselves. The gradient of the original clipped objective is indeed the negative of the gradient of the hinge loss objective. Their objective functions differ from each other by a constant[2], but their policy update rules derived by the gradient methods are exactly the same.

$$\frac{\partial}{\partial \theta} \min\{\rho_{s,a}(\theta)A^{\pi}(s,a), \text{clip}(\rho_{s,a}(\theta), 1 - \epsilon, 1 + \epsilon)A^{\pi}(s,a)\}$$

$$= -\frac{\partial}{\partial \theta} |A^{\pi}(s,a)| \, \ell(\text{sign}(A^{\pi}(s,a)), \rho_{s,a}(\theta) - 1, \epsilon). \quad (8)$$

Here, $\ell(y_i, f_{\theta}(x_i), \epsilon)$ is a hinge loss defined as

$$\max\{0, \epsilon - y_i \times f_{\theta}(x_i)\}, \quad (9)$$

where $\epsilon$ is the margin, $y_i \in \{-1, 1\}$ the label corresponding to the data $x_i$, and $f_{\theta}(x_i)$ serves as the binary classifier. Once $y_i f_{\theta}(x_i)$ is larger than the margin, $\ell(y_i, f_{\theta}(x_i), \epsilon)$ will equal zero, which reflects the sample clipping mechanism in PPO-clip. Note that hinge loss has been commonly used for large-margin classification, most notably for support vector machines (Freund & Schapire, 1999). From the above, in the tabular settings maximizing the objective function in (7) can be rewritten as minimizing the following loss function,

$$L(\theta) = \sum_{s \in \mathcal{S}} d_{\mu_0}^{\pi}(s) \sum_{a \in \mathcal{A}} \pi(a|s)|A^{\pi}(s,a)| \, \ell(\text{sign}(A^{\pi}(s,a)), \rho_{s,a}(\theta) - 1, \epsilon). \quad (10)$$

In practice, using the sample average to approximately maximize the above loss function as

$$L(\theta) \approx \hat{L}(\theta) = \sum_{(s,a) \in \mathcal{D}_{\pi}} \frac{|A^{\pi}(s,a)| \, \ell(\text{sign}(A^{\pi}(s,a)), \rho_{s,a}(\theta) - 1, \epsilon)}{|\mathcal{D}_{\pi}|}, \quad (11)$$

where $\mathcal{D}_{\pi}$ is a batch of samples drawn under $\pi$ and $|A^{\pi}(s,a)|$ can be interpreted as the weight or cost associated with each sample in large-margin classification.

**Remark 1.** For classification problems, if one is able to find a solution (i.e., classifier) that can classify all of the training data correctly, then the "costs" are relatively unimportant compared to the label. Therefore, in the hinge loss function form, this implies that the magnitude of advantage is not important. Note that we do not take the state-action pair with $A^{\pi}(s,a) = 0$ into account. As shown later in the experiments in Appendix E, algorithms with unweighted loss function also perform well. The "costs" in the cost-sensitive classification matter when the parameter space is not complete or when the hinge loss is not reduced to (nearly) zero. In those cases, one can consider it as a cost-sensitive classification problem and extend the analysis.

---

[2]Please see Appendix G for the detailed comparison of the two objectives.

**Hinge Policy Optimization.** Based on the the similarity of PPO-clip objective as that to minimize a hinge loss drawn by (Pi et al., 2020), we propose a new family of algorithms called Hinge Policy Optimization (HPO), where the general form of the loss function for policy improvement is

$$L(\theta) = \frac{1}{|\mathcal{D}|} \sum_{(s,a)\in\mathcal{D}} \text{weight} \times \ell(\text{label, classifier, margin}). \tag{12}$$

Different combinations of classifiers, margins, and weights lead to different loss functions, and hence represents different algorithms in this new HPO family.

## 2.3 HPO WITH AN ADAPTIVE MARGIN: ALGORITHM AND GLOBAL CONVERGENCE

In this subsection, we propose an HPO algorithm with an adaptive margin, called *HPO-AM*. In principle, HPO-AM resembles the PPO-clip algorithm, except for that we represent the objective function in a form of hinge loss, as described in (10) instead of (7), and that $\epsilon$ is dynamic for all states $s$ during the training process, as follows. Let $\hat{A}^\pi(s,a)$ denote the estimated advantage of a state-action pair $(s,a)$ under policy $\pi$. Then, we define

$$\epsilon_s^\pi := \alpha \cdot \min\Big\{1, \frac{\sum_{a\in I_s^{\pi-}}\pi(a|s)}{\sum_{a\in I_s^{\pi+}}\pi(a|s)}\Big\}, \ 0 < \alpha < 1, \ \forall s \in \mathcal{S}, \tag{13}$$

where $I_s^{\pi+} := \{a \in \mathcal{A}|\hat{A}^\pi(s,a) > 0\}$, and $I_s^{\pi-} := \{a \in \mathcal{A}|\hat{A}^\pi(s,a) < 0\}$. The notation $I_s^{\pi+}$ ($I_s^{\pi-}$) represents the set of actions of $s$ leading to positive (negative) estimated advantages. The adaptive clipping range is designed for the theoretical proof to "ensure the existence of an improved policy." The ratio in (13) reflects how much the current policy can be improved since HPO is designed to increase the probability of the actions with positive advantages and decrease those with negative advantages. Notably, the adaptive clipping range is in fact smaller than the constant clipping in the original PPO-clip as the adaptive clipping range is the constant $\alpha$ times a number no greater than 1.

Our HPO-AM is shown in Algorithm 1 below: Like PPO-clip, HPO-AM proceeds iteratively and, in each iteration $t$, runs an old policy (denoted by $\pi^{(t)}$ or $\pi(\theta^{(t)})$ parameterized by $\theta^{(t)}$) obtained from the previous iteration $t-1$, and then updates the new policy into $\pi^{(t+1)}$ for the next iteration. For simplicity, let $\rho_{s,a}^{(t)}(\theta)$, $\epsilon_s^{(t)}$, $I_s^{(t)+}$, and $I_s^{(t)-}$ respectively denote the above $\rho_{s,a}(\theta)$, $\epsilon_s^\pi$, $I_s^{\pi+}$, and $I_s^{\pi-}$ used during iteration $t$. Moreover, let $V^{(t)}(s)$, $Q^{(t)}(s,a)$, $A^{(t)}(s,a)$, and $\hat{A}^{(t)}(s,a)$ denote $V^{\pi^{(t)}}(s)$, $Q^{\pi^{(t)}}(s,a)$, $A^{\pi^{(t)}}(s,a)$, and $\hat{A}^{\pi^{(t)}}(s,a)$, respectively. Define the loss function of HPO-AM as

$$\hat{L}^{(t)}(\theta) = \sum_{\tau\in\mathcal{D}_t}\sum_{k=0}^{K_\tau}|\hat{A}^{(t)}(s_k,a_k)|\ell\big(\text{sign}(\hat{A}^{(t)}(s_k,a_k)), \rho_{s_k,a_k}^{(t)}(\theta) - 1, \epsilon_{s_k}^{(t)}\big). \tag{14}$$

In HPO-AM, we consider the weighted hinge loss with classifier $\rho_{s,a}(\theta) - 1$. In each iteration, HPO-AM updates the policy by the entropic mirror descent algorithm (EMDA) (Beck & Teboulle, 2003) to achieve a sufficiently small loss. Let $\delta_s^{(t)} := \min_{a\in I_s^{(t)+}\cup I_s^{(t)-}}|\hat{A}^{(t)}(s,a)|$ and set $\delta_s^{(t)} = \infty$ if $I_s^{(t)+}\cup I_s^{(t)-}$ is empty. To specify the termination condition of EMDA, define $\delta^{(t)} := \min_{s\in\mathcal{S}}\delta_s^{(t)}$ and $\epsilon^{(t)} := \min_{s\in\{s'\in\mathcal{S}|\epsilon_{s'}^{(t)}>0\}}\epsilon_s^{(t)}$, and choose the threshold as $(1-\zeta)\delta^{(t)}\epsilon^{(t)}$, with $\zeta \in (0,1)$.

**Remark 2.** As shown in Algorithm 1, EMDA is applied to find a proper $\theta^{(t+1)}$ that achieves a sufficiently small loss $\hat{L}^{(t)}(\theta)$. While there are alternative ways to minimize the loss $\hat{L}^{(t)}(\theta)$ over $\Delta(\mathcal{A})^{|\mathcal{S}|}$ (e.g., the projected subgradient method), we leverage the exponentiated gradient scheme of EMDA to ensure that $\pi^{(t)}$ remains strictly positive for all state-action pairs in each iteration $t$.

Before obtaining our main theorem, we make the following assumptions.

**Assumption 3** (Training data and strict positivity of distributions). *Suppose that the sampled training data set during each iteration of HPO-AM is sufficiently large to contain all possible state-action pairs. Strict positivity of the initial state distribution $\mu_0$ and the initial policy $\pi^{(0)}$ is a necessary condition for this assumption.*

**Assumption 4** (Correctness of the sign of the advantage function). *We assume that at each iteration $t$, for each state-action pair, the sign of the estimated advantage is the same as that of the true advantage, i.e., $\text{sign}(\hat{A}^{(t)}(s,a)) = \text{sign}(A^{(t)}(s,a))$, for all $(s,a)$.*

| **Algorithm 1:** HPO with an adaptive margin | **Algorithm 2:** EMDA($\mathcal{L}(\theta), \{\eta_k\}, \xi, \theta_{\text{init}}$) |
|---|---|
| **Result:** Learned policy $\pi^{(\infty)}$ | **Result:** Learned parameter $\widetilde{\theta}$ |
| 1 Initialize policy $\pi^{(0)} = \pi(\theta^{(0)})$, initial state distribution $\mu_0$, $\alpha \in (0, 1)$, step sizes of EMDA $\{\eta_k\}$, threshold $\zeta \in (0, 1)$; | 1 **Input:** Objective $\mathcal{L}(\theta)$, step sizes $\{\eta_k\}$, threshold $\xi$, and initial parameter $\theta_{\text{init}}$; |
| 2 **for** $t = 0, 1, \cdots$ **do** | 2 Initialize $\widetilde{\theta}^{(1)} = \theta_{\text{init}}$, $\widetilde{\theta} = \theta_{\text{init}}$, and $k = 1$; |
| 3 $\quad$ Collect a set of trajectories $\tau \in \mathcal{D}_t$ under policy $\pi^{(t)} = \pi(\theta^{(t)})$; | 3 **while** $\mathcal{L}(\widetilde{\theta}) > \xi$ **do** |
| 4 $\quad$ Estimate $\hat{A}^{(t)}$ by any policy evaluation algorithms; | 4 $\quad$ **for** *each state $s$* **do** |
| 5 $\quad$ Compute $\hat{L}^{(t)}(\theta), \delta^{(t)}, \epsilon^{(t)}$ based on $\hat{A}^{(t)}$ and the collected samples in $\mathcal{D}_t$; | 5 $\quad\quad$ Find $g_{s,a} = \frac{\partial \mathcal{L}(\theta)}{\partial \theta_{s,a}}\big|_{\theta = \widetilde{\theta}^{(k)}}$, for each $a$; |
| 6 $\quad$ Update the policy by $\theta^{(t+1)} = \text{EMDA}(\hat{L}^{(t)}(\theta), \{\eta_k\}, (1 - \zeta)\delta^{(t)}\epsilon^{(t)}, \theta^{(t)})$ | 6 $\quad\quad$ Let $w_s = (e^{-\eta_k g_{s,1}}, \cdots, e^{-\eta_k g_{s,|\mathcal{A}|}})$; |
| 7 **end** | 7 $\quad\quad$ $\widetilde{\theta}_s^{(k+1)} = \frac{1}{\langle w_s, \widetilde{\theta}_s^{(k)} \rangle}(w_s \circ \widetilde{\theta}_s^{(k)})$; |
| | 8 $\quad$ **end** |
| | 9 $\quad$ $\widetilde{\theta} = \arg\min_{\theta \in \{\widetilde{\theta}^{(i)}: 1 \le i \le k+1\}} \mathcal{L}(\theta)$; |
| | 10 $\quad$ Increment $k$ by 1; |
| | 11 **end** |

**Remark 3.** As pointed out by Assumption 4, the global convergence of HPO requires only the *correct signs* of the advantage values, as will be shown in Theorem 1 below. This is one major difference from the recent global convergence results of policy gradient methods (Agarwal et al., 2019; Bhandari & Russo, 2019; Mei et al., 2020), which require the true policy gradient and hence the true advantage function. To address the potentially incorrect signs of the estimated advantage values, one could consider the "label-robust" version of HPO by leveraging the loss functions in the robust classification literature, e.g., (Bertsimas et al., 2019; Natarajan et al., 2013). As this paper is meant to take the first step toward establishing the theoretical foundation of the proposed new policy optimization framework and PPO-clip, we leverage Assumption 3 in order to rigorously establish the global convergence property of HPO.

**Theorem 1** (Global convergence of HPO-AM). *Suppose the step size of EMDA is diminishing and non-summable. Under HPO-AM, we have $V^{(t)}(s) \to V^*(s)$ as $t \to \infty$, $\forall s \in \mathcal{S}$.*

The proof of Theorem 1 is provided in the next section. Notably, the main difference between HPO-AM and PPO-clip is the choice of margin, and HPO-AM is ensured to monotonically increase all state values, not just the expected total reward as in PPO-KL and TRPO. Moreover, Theorem 1 ensures that all of the state values converge to the optimal value eventually.

Inspired by (Agarwal et al., 2020), based on the limits $A^{(\infty)}(s, a)$ which can be shown to exist via Theorem 1, we divide all actions into the following sets for each state $s$:

$$I_s^+ := \{a \in \mathcal{A} | A^{(\infty)}(s, a) > 0\}, \tag{15}$$

$$I_s^0 := \{a \in \mathcal{A} | A^{(\infty)}(s, a) = 0\}, \tag{16}$$

$$I_s^- := \{a \in \mathcal{A} | A^{(\infty)}(s, a) < 0\}. \tag{17}$$

Note that $\pi^{(t)}$ converges to an optimal policy if $I_s^+$ is empty for all $s$. As a result, $I_s^+$ will play an important role in the proof of Theorem 1 in Section 3.2. Theorem 1 is followed by the two notable Corollaries 1-2, and their proofs are given in Appendix B.

**Corollary 1.** *Consider state $s \in \mathcal{S}$. For each $a \in I_s^-$, the probability for our agent to take action $a$ under state $s$ converges to zero. For $a \in I_s^0$, the probability sum of taking these actions under state $s$ converges to one. That is, $\pi^{(t)}(a|s) \to 0$ for all $a \in I_s^-$ and $\sum_{a \in I_s^0} \pi^{(t)}(a|s) \to 1$ as $t \to \infty$.*

**Corollary 2.** *Consider state $s \in \mathcal{S}$, if there is only one action $a \in I_s^0$, i.e., the optimal policy is deterministic, then we have $\epsilon_s^{(t)} \to 0$ as $t \to \infty$.*

**Remark 4** (Multiple optimal actions)**.** In the case that a state has one or more actions with the largest Q-value, any policy that assigns nonzero probability only to these actions with one in total is an optimal policy. From Corollary 1, we know that the probability sum of actions belong to $I_s^0$ approaches one. As a result, we presume that the actions described above form the set $I_s^0$.

**Remark 5** (Label quality). In Theorem 1, we obtain a similar state-wise convergence result as Theorem 5.1 in (Agarwal et al., 2020), which proves asymptotic state-wise global convergence of policy gradient for softmax parameterization. In (Agarwal et al., 2020), for the convergence result to be true, it requires true gradient, which includes true advantage values. In addition, the calculation for true gradient is an expectation over state visitation frequency, so it needs to follow a specific distribution and sum over every state-action pair. In contrast, as stated in Remark 1 and Assumptions 3-4, our method only requires *true advantage signs* and visiting all state action pairs, and there is no need to approximate the state visitation frequency. Obviously, the requirement of accurate advantage estimate is more demanding than that of accurate advantage signs.

## 3 PROOF OF GLOBAL CONVERGENCE OF HPO-AM

### 3.1 SUPPORTING LEMMAS FOR THE PROOF OF THEOREM 1

In this section, we present the important properties of HPO-AM, which would serve as useful lemmas for proving Theorem 1. Define the *state-wise* loss function for each state $s$ in each iteration $t$ as

$$\hat{L}_s^{(t)}(\theta_s) = \sum_{a \in \mathcal{A}} |\hat{A}^{(t)}(s, a)| \, \ell(\text{sign}(\hat{A}^{(t)}(s, a)), \rho_{s,a}(\theta_s) - 1, \epsilon_s^{(t)}). \tag{18}$$

Based on Assumption 2, where $\theta_s \in \Delta(\mathcal{A})$, we can use $\rho_{s,a}(\theta_s)$ to represent $\rho_{s,a}(\theta)$ and further construct the above function of $\theta_s$. Regardless of the size of the state space, the loss function can be viewed as optimizing policies on a per-state basis. The reason for this lies in that the definition of state-wise loss function is valid as long as the policy parameterization is "separable by state." That is, the policy parameters under different states are independent, e.g., the direct tabular parameterization and the softmax parameterization. As a result, we know $\hat{L}^{(t)}(\theta) = 0$ if and only if $\hat{L}_s^{(t)}(\theta_s) = 0$ for all $s \in \mathcal{S}$. Next, we introduce the supporting lemmas, whose proofs are in Appendix A.

**Lemma 1** (Strict improvement and strict positivity of policy under HPO-AM). *Suppose $\pi^{(t)}$ is strictly positive in all state-action pairs, i.e., $\pi^{(t)}(a|s) > 0$, for all $(s, a)$. Under HPO-AM, if EMDA takes diminishing non-summable step sizes, i.e., $\eta_k \to 0$ and $\sum_{k=1}^{\infty} \eta_k = \infty$, then $\pi^{(t+1)}$ satisfies that $\pi^{(t+1)} > \pi^{(t)}$ and $\pi^{(t+1)}(a|s) > 0$, for all $(s, a)$.*

**Lemma 2** (Monotonicity in $\pi^{(t)}(a|s)$). *There exist $T_0$ and $T_1$ such that for all $a \in I_s^+$, $\pi^{(t)}(a|s)$ is strictly increasing for $t > T_0$; for all $a \in I_s^-$, $\pi^{(t)}(a|s)$ is strictly decreasing for $t > T_1$.*

**Lemma 3** (Lower Bound of $\epsilon_s^{(t)}$). *If $I_s^+$ is not an empty set, then there exists a positive constant $c$ such that $\epsilon_s^{(t)}$ is lower bounded by $c$, for all $t > T_1$.*

**Lemma 4.** $\sum_{a \in I_s^+ \cup I_s^-} \epsilon_s^{(t)} \pi^{(t)}(a|s) |A^{(t)}(s, a)| \to 0$ as $t \to \infty$.

### 3.2 PROOF OF THEOREM 1

We first prove that each state value increases monotonically. Then, under the bounded reward setting, we have that each state value is bounded and thus converges. For convergence to the global optimal, we prove that the algorithm won't stop until there is no action with positive advantage. The same analytical framework has also been used by (Agarwal et al., 2020) to establish the global convergence of the vanilla policy gradient method. Despite the high-level resemblance, we reach the above three proof steps in a totally different way as the two approaches are fundamentally different.

From Lemma 1, we know that in each iteration $t$, $\pi^{(t+1)}$ and $\pi^{(t)}$ satisfy the sufficient condition for state-wise policy improvement provided by Proposition 2. This implies that the sequence of state values is point-wise monotonically increasing, i.e., $V^{(t+1)}(s) \geq V^{(t)}(s)$, $\forall s \in \mathcal{S}$. Moreover, by Assumption 1 and the discounted setting, we have $-\frac{R}{1-\gamma} \leq V^{(t)}(s) \leq \frac{R}{1-\gamma}$. The above monotone increasing property and boundedness imply convergence, i.e., $V^{(t)}(s) \to V^{(\infty)}(s)$. Similarly, it can be derived that $Q^{(t)}(s, a) \to Q^{(\infty)}(s, a)$, and thus $A^{(t)}(s, a) \to A^{(\infty)}(s, a)$. The above sets $I_s^+$, $I_s^0$ and $I_s^-$ are well-defined, based on the limit $A^{(\infty)}(s, a)$.

To establish that $\pi^{(t)}$ converges to an optimal policy, it is sufficient to show that $I_s^+$ is an empty set for all $s$. The proof proceeds by contradiction as follows: Suppose $I_s^+$ is non-empty with $a_+ \in I_s^+$. From Lemma 2, we have that $\pi^{(t)}(a_+|s)$ is bounded below by $\pi^{(T_0)}(a_+|s)$ for some $T_0 \in \mathbb{N}$ for all $t > T_0$. From Lemma 3, there exists a positive lower bound of $\epsilon_s^{(t)}$. However, Lemma 4 shows that convergence of $V^{(t)}(s)$ implies that either $\epsilon_s^{(t)}$ converges to zero or the sum of probability mass over all actions in $I_s^+ \cup I_s^-$ converges to zero. This leads to a contraction, and thus completes the proof.

## 4    HPO-AM WITH ALTERNATIVE CLASSIFIERS

In the previous sections, we propose the HPO update scheme and prove the global convergence for a particular case, HPO-AM, in Theorem 1. In this section, we go beyond HPO-AM as follows:

**HPO-AM with Alternative Classifiers.** Notably, the HPO update scheme can be readily extended to various other classifiers. In the following Theorem 2, we provide a further result that global convergence also holds for HPO-AM with some other classifiers. The proof is given in Appendix C.

**Theorem 2.** *Theorem 1 is also satisfied for HPO-AM with the following classifiers and the corresponding margins with $0 < \alpha < 1$:*

*(i) HPO-AM with classifier $\log(\pi_\theta(a|s)) - \log(\pi(a|s))$ (HPO-AM-log): Let*

$$\epsilon_s^\pi = \log(1 + \alpha \cdot \min\{1, \frac{\sum_{a \in I_s^{\pi-}} \pi(a|s)}{\sum_{a \in I_s^{\pi+}} \pi(a|s)}\}). \tag{19}$$

*(ii) HPO-AM with classifier $\sqrt{\rho_{s,a}(\theta)} - 1$ (HPO-AM-root): Let*

$$\epsilon_s^\pi = \sqrt{1 + \alpha \cdot \min\{1, \frac{\sum_{a \in I_s^{\pi-}} \pi(a|s)}{\sum_{a \in I_s^{\pi+}} \pi(a|s)}\}} - 1. \tag{20}$$

**Remark 6.** The classifiers in Theorem 2 also leverage the probability ratio $\rho_{s,a}(\theta)$, but in a way different from that in HPO-AM in (14). Given the generality of the HPO framework, one can design HPO-AM with other customized classifiers that either does not directly involve the probability ratio or considers the power of the ratio to be larger than one, such as $\pi_\theta(a|s) - \pi(a|s)$ (called HPO-AM-sub) or $(\rho_{s,a}(\theta))^2 - 1$ (called HPO-AM-square). It remains an interesting open problem how to establish global convergence for HPO with these classifiers. We provide more detailed explanations for the above two unproven classifiers in Appendix D.

## 5    PRACTICAL IMPLEMENTATION AND EXPERIMENTAL RESULTS

Recall that HPO-AM improves the policy by minimizing the loss $\hat{L}^{(t)}(\theta)$ in (14). Therefore, like PPO-clip, in practice HPO-AM can be readily combined with neural network parameterization and end-to-end training. We implement HPO-AM and its variants on top of the open-source RL Baselines 3 Zoo framework (Raffin, 2020). We empirically evaluate the HPO-AM algorithms in MinAtar (Young & Tian, 2019), a modified version of Atari games for efficient evaluation, and conduct an ablation study to better understand the candidate design choices of HPO. Due to the space limitation, the ablation study is provided in Appendix E. We compare HPO-AM variants with PPO-clip, A2C, and Rainbow in Asterix, Breakout and Freeway with 5 million training steps under 5 random seeds. For PPO-clip and A2C, we use the tuned hyperparameters suggested by Stable Baselines 3, and we disable the entropy regularization for a fair comparison. For Rainbow, we use the open-source implementation from (Obando-Ceron & Castro, 2021).

Figure 1 shows the training curves of the HPO algorithms and multiple popular benchmark RL methods in three different games. The results of A2C and Rainbow are consistent with their performance in the benchmark (Obando-Ceron & Castro, 2021). In (Kaiser et al., 2020), for Atari games, Rainbow is better than PPO-clip in Asterix, but worse in Breakout and Freeway, which is similar to our results in MinAtar. Figure 1 indicates that each classifier has its own strengths. Notably, HPO-AM and HPO-AM-root achieve comparable or better performance than PPO-clip in the three environments. HPO-AM-log gets similar rewards as PPO-clip in Breakout and Freeway, but it gets fewer rewards than PPO-clip in Asterix. HPO-AM-square is close to A2C in Asterix and Breakout, but its performance is on par with PPO-clip in Freeway. Overall, the above results demonstrate that the HPO algorithms are indeed applicable in high-dimensional tasks.

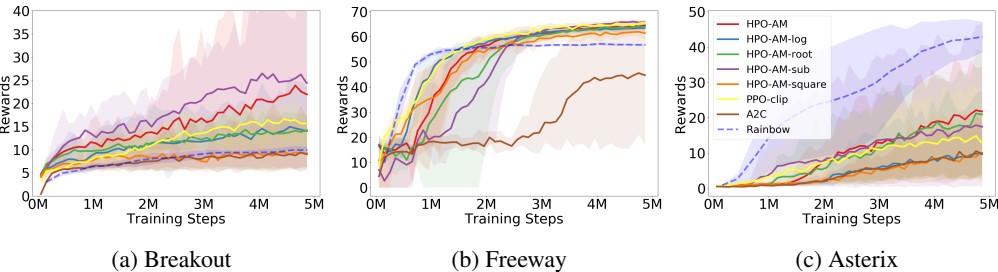

| (a) Breakout | (b) Freeway | (c) Asterix |

Figure 1: Experimental results in the MinAtar environments.

# 6 RELATED WORK

**Global Convergence of Policy Gradient Methods.** One related line of recent research is on the global convergence of the policy gradient methods. (Agarwal et al., 2019; 2020) establishes global convergence results of various policy gradient approaches, including the vanilla policy gradient (with both tabular and softmax policy parametrizations) and the natural policy gradient method (with a softmax policy parametrization). Concurrently, (Bhandari & Russo, 2019) provides an alternative analysis of global optimality of the policy gradient method. (Wang et al., 2019) provides the global optimality guarantees for both the vanilla policy gradient and natural policy gradient methods under the overparameterized two-layer neural parameterization. (Mei et al., 2020) establishes the convergence rates of both vanilla softmax policy gradient and the entropy-regularized policy gradient. (Liu et al., 2020) further establishes the global convergence rates of various variance-reduced policy gradient methods. Inspired by the analysis of (Agarwal et al., 2019), we establish the global convergence result of the proposed HPO-AM algorithms.

**Global Convergence of TRPO and PPO.** Regarding TRPO, (Shani et al., 2020) presents the global convergence rates of an adaptive TRPO, which is established by connecting TRPO and the mirror descent method. (Liu et al., 2019) proves global convergence in expected total reward for a neural variant of PPO with adaptive Kullback-Leibler penalty (PPO-KL). To the best of our knowledge, (Liu et al., 2019) appears to be the only global convergence result for PPO-KL. By contrast, our focus is PPO-clip. Given the salient algorithmic difference between PPO-KL and PPO-clip, there remains no proof of global convergence to an optimal policy for PPO with a clipped objective. In this paper, we rigorously provide the first global convergence guarantee for a variant of PPO-clip.

**RL as Classification.** Regarding the general idea of casting RL as a classification problem, it has been investigated by the existing literature (Lagoudakis & Parr, 2003; Lazaric et al., 2010; Farahmand et al., 2014), which view the one-step greedy update (e.g. in Q-learning) as a binary classification problem. However, a major difference is the labeling: Classification-based Approximate Policy Iteration labels the action with the largest Q value as positive; HPO labels the actions with positive advantage as positive. Despite the high-level resemblance, our paper is fundamentally different from the prior works (Lagoudakis & Parr, 2003; Lazaric et al., 2010; Farahmand et al., 2014) as our paper is meant to study the theoretical foundation of PPO-clip, from the perspective of HPO.

# 7 CONCLUDING REMARKS

In this paper, we propose to rethink policy optimization methods from the perspective of state-wise policy improvement. Specifically, we propose a classification-based policy update scheme leveraging hinge loss for policy improvement, and thereby propose a new family of algorithms called Hinge Policy Optimization (HPO). This paper proves global convergence of HPO with an adaptive margin (HPO-AM), which can be viewed as a PPO-clip algorithm with an adaptive clipping range. Tabular parameterization enables us to perform policy updates for each state separately; viewing each state-action pair as a training sample allows checking state-wise policy improvement, and the adaptive margin ensures the existence of improved policy. We also conduct experiments in both tabular and more complex environments for several variants of HPO-AM algorithms equipped with different classifiers and margins to corroborate their convergence behavior and performance.

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

APPENDIX

# A    PROOFS OF THE SUPPORTING LEMMAS FOR THEOREM 1

## A.1    ADDITIONAL SUPPORTING LEMMAS

**Lemma 5** (Loss value and policy improvement). *Suppose that $\pi^{(t)}(a|s) > 0$, for all state-action pairs $(s, a)$. Given any $\zeta \in (0, 1)$, if $\hat{L}_s^{(t)}(\theta_s) \le (1-\zeta)\delta_s^{(t)}\epsilon_s^{(t)}$ for all $s \in \mathcal{S}$, then we have $\pi_\theta > \pi^{(t)}$.*

*Proof.* Recall that

$$\hat{L}_s^{(t)}(\theta_s) = \sum_{a \in \mathcal{A}} |\hat{A}^{(t)}(s,a)| \, \ell(\text{sign}(\hat{A}^{(t)}(s,a)), \rho_{s,a}(\theta_s) - 1, \epsilon_s^{(t)}) \tag{21}$$

$$= \sum_{a \in \mathcal{A}} |\hat{A}^{(t)}(s,a)| \, \ell(\text{sign}(A^{(t)}(s,a)), \rho_{s,a}(\theta_s) - 1, \epsilon_s^{(t)}) \tag{22}$$

$$\ge \sum_{a \in \mathcal{A}} \delta_s^{(t)} \max\{0, \epsilon_s^{(t)} - \text{sign}(A^{(t)}(s,a))(\rho_{s,a}(\theta_s) - 1)\}, \tag{23}$$

where (22) holds by Assumption 4, and (23) holds by the definition of $\delta_s^{(t)}$ and the non-negativity of hinge loss. Then, our goal is to show the following claim: for each state $s$, $\hat{L}_s^{(t)}(\theta_s) \le (1-\zeta)\delta_s^{(t)}\epsilon_s^{(t)}$ implies that

$$\pi_\theta(a|s) \ge (1 + \zeta\epsilon_s^{(t)})\pi^{(t)}(a|s), \quad a \in I_s^{(t)+}, \tag{24}$$

$$\pi_\theta(a|s) \le (1 - \zeta\epsilon_s^{(t)})\pi^{(t)}(a|s), \quad a \in I_s^{(t)-}. \tag{25}$$

We can prove the above claim by contradiction. If there exists an action $a \in I_s^{(t)+}$ such that $\pi_\theta(a|s) < (1 + \zeta\epsilon_s^{(t)})\pi^{(t)}(a|s)$, then we know $\hat{L}_s^{(t)}(\theta_s) > (1 - \zeta)\delta_s^{(t)}\epsilon_s^{(t)}$ by (23). Similarly, if there exists an action $a \in I_s^{(t)-}$ such that $\pi_\theta(a|s) > (1 - \zeta\epsilon_s^{(t)})\pi^{(t)}(a|s)$, then we know $\hat{L}_s^{(t)}(\theta_s) > (1 - \zeta)\delta_s^{(t)}\epsilon_s^{(t)}$ by (23). Therefore, by (24)-(25) and the sufficient condition for policy improvement in Proposition 1, we know $\pi_\theta \ge \pi^{(t)}$. $\qquad\square$

**Lemma 6** (Existence of a policy with zero loss). *Given that the tabular policy parameterization is complete, i.e., any stochastic policy can be represented in the class, in each iteration $t$, we can always find a policy $\pi_\theta$ such that $\hat{L}_s^{(t)}(\theta_s) = 0$, for all $s \in \mathcal{S}$.*

*Proof.* Recall that $I_s^{(t)+} := \{a \in \mathcal{A} | \hat{A}^{(t)}(s,a) > 0\}$ and $I_s^{(t)-} := \{a \in \mathcal{A} | \hat{A}^{(t)}(s,a) < 0\}$. Notice that we ignore the actions with zero advantage. We will prove that there always exists a policy $\pi_\theta$ such that

$$\text{sign}(\hat{A}^{(t)}(s,a))(\rho_{s,a}(\theta) - 1) \ge \epsilon_s^{(t)} \tag{26}$$

for all $a \in I_s^{(t)+} \cup I_s^{(t)-}$, i.e., actions such that $A^\pi(s,a) \ne 0$, for all $s \in \mathcal{S}$.

For state $s$, take

$$\pi_\theta(a|s) = \theta_{s,a} = \left(1 + \alpha \cdot \frac{\sum_{a \in I_s^{(t)-}} \pi(a|s)}{\sum_{a \in I_s^{(t)+}} \pi(a|s)}\right)\pi(a|s), \forall a \in I_s^{(t)+}, \tag{27}$$

$$\pi_\theta(a|s) = \theta_{s,a} = (1 - \alpha)\pi(a|s), \forall a \in I_s^{(t)-}. \tag{28}$$

We consider two cases, depending on whether the probability sum of actions with positive advantage is bigger than those with negative advantage, under the current state.

**Case 1.** If $\frac{\sum_{a \in I_s^{(t)-}} \pi(a|s)}{\sum_{a \in I_s^{(t)+}} \pi(a|s)} \ge 1$, then $\epsilon_s^{(t)} = \alpha$.

For $a \in I_s^{(t)+}$, we have

$$\pi_\theta(a|s) = \left(1 + \alpha \cdot \frac{\sum_{a \in I_s^{(t)-}} \pi(a|s)}{\sum_{a \in I_s^{(t)+}} \pi(a|s)}\right)\pi(a|s)$$

$$\geq (1+\alpha)\pi(a|s) = (1+\epsilon_s^{(t)})\pi(a|s). \tag{29}$$

On the other hand, for $a \in I_s^{(t)-}$, we have

$$\pi_\theta(a|s) = (1-\alpha)\pi(a|s) = (1-\epsilon_s^{(t)})\pi(a|s). \tag{30}$$

**Case 2.** If $\frac{\sum_{a \in I_s^{(t)-}} \pi(a|s)}{\sum_{a \in I_s^{(t)+}} \pi(a|s)} < 1$, then $\epsilon_s^{(t)} = \alpha \cdot \frac{\sum_{a \in I_s^{(t)-}} \pi(a|s)}{\sum_{a \in I_s^{(t)+}} \pi(a|s)}$.

For $a \in I_s^{(t)+}$, we have

$$\pi_\theta(a|s) = \left(1 + \alpha \cdot \frac{\sum_{a \in I_s^{(t)-}} \pi(a|s)}{\sum_{a \in I_s^{(t)+}} \pi(a|s)}\right)\pi(a|s) = (1+\epsilon_s^{(t)})\pi(a|s). \tag{31}$$

Similarly, for $a \in I_s^{(t)-}$, we know

$$\pi_\theta(a|s) = (1-\alpha)\pi(a|s) \leq \left(1 - \alpha \cdot \frac{\sum_{a \in I_s^{(t)-}} \pi(a|s)}{\sum_{a \in I_s^{(t)+}} \pi(a|s)}\right)\pi(a|s)$$

$$= (1-\epsilon_s^{(t)})\pi(a|s). \tag{32}$$

Therefore, in both cases we have $\pi_\theta$ satisfy (26) for all $a \in I_s^{(t)+} \cup I_s^{(t)-}$ and for all $s \in \mathcal{S}$, i.e., $\hat{L}^{(t)}(\theta) = 0$. $\qquad\square$

## A.2 PROOF OF LEMMA 1

For convenience, we restate Lemma 1 as follows.

**Lemma** (Strict improvement and strict positivity of policy under HPO-AM). *Suppose $\pi^{(t)}$ is strictly positive in all state-action pairs, i.e., $\pi^{(t)}(a|s) > 0$, for all $(s,a)$. Under HPO-AM, if EMDA takes diminishing non-summable step sizes, i.e., $\eta_k \to 0$ and $\sum_{k=1}^{\infty} \eta_k = \infty$, then $\pi^{(t+1)}$ satisfies that $\pi^{(t+1)} > \pi^{(t)}$ and $\pi^{(t+1)}(a|s) > 0$, for all $(s,a)$.*

*Proof.* We start by showing that the objective loss function $\hat{L}^{(t)}(\theta)$ is convex and Lipschitz continuous. From (18), we have

$$\hat{L}_s^{(t)}(\theta_s) = \sum_{a \in \mathcal{A}} |\hat{A}^{(t)}(s,a)| \, \ell(\text{sign}(\hat{A}^{(t)}(s,a)), \rho_{s,a}(\theta_s) - 1, \epsilon_s^{(t)}), \tag{33}$$

$$= \sum_{a \in \mathcal{A}} |\hat{A}^{(t)}(s,a)| \max\{0, \epsilon_s^{(t)} - \text{sign}(\hat{A}^{(t)}(s,a))(\rho_{s,a}(\theta_s) - 1)\}. \tag{34}$$

By the fact that $\rho_{s,a}(\theta_s) - 1$ is linear and hinge loss is convex (Rosasco et al., 2004), their composition $\ell(\text{sign}(\hat{A}^\pi(s,a)), \rho_{s,a}(\theta_s) - 1, \epsilon_s^{(t)})$ preserves convexity, for each $a \in \mathcal{A}$. Being the nonnegative weighted sum of the above convex functions, $\hat{L}_s^{(t)}(\theta_s)$ is also convex. Hence, $\hat{L}^{(t)}(\theta)$ is also convex.

For Lipschitz continuity, it follows from the fact that for any $\theta, \bar{\theta} \in \Delta(\mathcal{A})^{|\mathcal{S}|}$,

$$|\max\{0, \epsilon_s - \text{sign}(\hat{A}^{(t)}(s,a))(\rho_{s,a}(\theta_s) - 1)\} - \max\{0, \epsilon_s - \text{sign}(\hat{A}^{(t)}(s,a))(\rho_{s,a}(\bar{\theta}_s) - 1)\}|$$

$$\leq |(\epsilon_s^{(t)} - \text{sign}(\hat{A}^{(t)}(s,a))(\rho_{s,a}(\theta_s) - 1)) - (\epsilon_s^{(t)} - \text{sign}(\hat{A}^{(t)}(s,a))(\rho_{s,a}(\bar{\theta}_s) - 1))| \tag{35}$$

$$= |\rho_{s,a}(\theta_s) - \rho_{s,a}(\bar{\theta}_s)| \tag{36}$$

$$= \frac{1}{\pi^{(t)}(a|s)} |\pi_\theta(a|s) - \pi_{\bar{\theta}}(a|s)| \tag{37}$$

$$= \frac{1}{\pi^{(t)}(a|s)} |\theta_{s,a} - \bar{\theta}_{s,a}| \tag{38}$$

$$\leq \frac{1}{\pi^{(t)}(a|s)} \|\theta_s - \bar{\theta}_s\|. \tag{39}$$

Thus, $|\hat{A}^{(t)}(s,a)| \max\{0, \epsilon_s^{(t)} - \text{sign}(\hat{A}^{(t)}(s,a))(\rho_{s,a}(\theta_s) - 1)\}$ is Lipschitz continuous with a Lipschitz constant $\frac{|\hat{A}^{(t)}(s,a)|}{\pi^{(t)}(a|s)}$, for each $a \in \mathcal{A}$. As a result, the summation $\hat{L}_s^{(t)}(\theta_s)$ over all actions is $M$-Lipschitz continuous with $M = \frac{\frac{2R}{1-\gamma}}{\min\{\pi^{(t)}(a|s)\}} < \infty$ (since $\pi^{(t)}(a|s) > 0$ for all $s, a$). Hence, $\hat{L}^{(t)}(\theta)$ is Lipschitz continuous as well.

Therefore, under a diminishing non-summable step size, the convergence of $\hat{L}^{(t)}(\theta)$ to its minimum, which is zero by Lemma 6, follows directly from the standard convergence result of EMDA shown by (Beck & Teboulle, 2003, Theorem 4.1). Moreover, by the termination condition of EMDA specified in Algorithm 1, we know that the $\theta$ returned by EMDA must satisfy $\hat{L}^{(t)}(\theta) \leq (1-\zeta)\delta^{(t)}\epsilon^{(t)}$, which implies that $\hat{L}_s^{(t)}(\theta) \leq (1-\zeta)\delta_s^{(t)}\epsilon_s^{(t)}$, for every state $s \in \mathcal{S}$. Hence, by Lemma 5, we have $\pi^{(t+1)} > \pi^{(t)}$.

Next, by the strict positivity of $\pi^{(t)}$, we also have $\delta_s^{(t)} > 0$ and $\epsilon_s^{(t)} > 0$, for all $s$. Then, by the termination condition of EMDA specified in Algorithm 1 and the convergence of EMDA, we know that under HPO-AM, EMDA would terminate in finitely many iterations (i.e., finite $k$ in Algorithm 2), for any $t$. Therefore, it is easy to verify that $\pi^{(t+1)}(a|s) > 0$ for all $(s,a)$ by the exponentiated gradient update scheme of EMDA and the strict positivity of $\pi^{(t)}$. $\qquad \square$

### A.3 PROOF OF LEMMA 2

For ease of exposition, we restate Lemma 2 as follows.

**Lemma** (Monotonicity in $\pi^{(t)}(a|s)$). *There exist $T_0$ and $T_1$ such that for all $a \in I_s^+$, $\pi^{(t)}(a|s)$ is strictly increasing for $t > T_0$; for all $a \in I_s^-$, $\pi^{(t)}(a|s)$ is strictly decreasing for $t > T_1$.*

*Proof.* Consider $a \in I_s^+$. Since $A^{(\infty)}(s,a) > 0$, there exists a $T_0 \in \mathbb{N}$ such that $A^{(t)}(s,a) > 0$ for all $t > T_0$. This implies that

$$\pi^{(t+1)}(a|s) - \pi^{(t)}(a|s) > \epsilon_s^{(t)}\pi^{(t)}(a|s) > 0, \tag{40}$$

i.e., $\pi^{(t)}(a|s)$ is strictly increasing for $t > T_0$.

The proof of the second claim for $a \in I_s^-$ is similar. $\qquad \square$

### A.4 PROOF OF LEMMA 3

For ease of exposition, we restate Lemma 3 as follows.

**Lemma** (Lower Bound of $\epsilon_s^{(t)}$). *If $I_s^+$ is not an empty set, then there exists a positive constant $c$ such that $\epsilon_s^{(t)}$ is lower bounded by $c$, for all $t > T_1$.*

*Proof.* From the fact that

$$\sum_{a \in \mathcal{A}} \pi^{(\infty)}(a|s) A^{(\infty)}(s, a) = 0; \tag{41}$$

$$\sum_{a \in I_s^0} \pi^{(\infty)}(a|s) A^{(\infty)}(s, a) = 0, \tag{42}$$

we have

$$\sum_{a \in I_s^+} \pi^{(\infty)}(a|s) A^{(\infty)}(s, a) = \sum_{a \in I_s^-} \pi^{(\infty)}(a|s) |A^{(\infty)}(s, a)|. \tag{43}$$

By Assumption 1, we have $|V^{(\infty)}(s)| \le \frac{R}{1-\gamma}$, $|Q^{(\infty)}(s, a)| \le \frac{R}{1-\gamma}$ and thus $|A^{(\infty)}(s, a)| = |Q^{(\infty)}(s, a) - V^{(\infty)}(s)| \le \frac{2R}{1-\gamma}$. Therefore, by Hölder's inequality, we have

$$\sum_{a \in I_s^-} \pi^{(\infty)}(a|s) \ge \frac{\sum_{a \in I_s^+} \pi^{(\infty)}(a|s) A^{(\infty)}(s, a)}{\frac{2R}{1-\gamma}}. \tag{44}$$

Define $\Delta := \min_{a \in I_s^+} A^{(\infty)}(s, a) > 0$ and $p := \sum_{a \in I_s^+} \pi^{(\infty)}(a|s)$. Since we know that $p = \sum_{a \in I_s^+} \pi^{(\infty)}(a|s) > \sum_{a \in I_s^+} \pi^{(T_0)}(a|s)$ by Lemma 2, we have $p > 0$. Moreover, by Lemma 2, we have $I_s^{(t)-} \supseteq I_s^-$ and $\pi^{(t+1)}(a|s) < \pi^{(t)}(a|s)$ for all $a \in I_s^-$ and $t > T_1$. This implies

$$\sum_{a \in I_s^{(t)-}} \pi^{(t)}(a|s) > \sum_{a \in I_s^-} \pi^{(t)}(a|s) > \sum_{a \in I_s^-} \pi^{(\infty)}(a|s) \ge \frac{p\Delta}{\frac{2R}{1-\gamma}} \tag{45}$$

for all $t > T_1$.

Therefore, by the definition of $\epsilon_s^{(t)}$, we have

$$\epsilon_s^{(t)} = \alpha \cdot \min\{1, \frac{\sum_{a \in I_s^{(t)-}} \pi^{(t)}(a|s)}{\sum_{a \in I_s^{(t)+}} \pi^{(t)}(a|s)}\} \tag{46}$$

$$\ge \alpha \cdot \frac{\sum_{a \in I_s^{(t)-}} \pi^{(t)}(a|s)}{\sum_{a \in I_s^{(t)+}} \pi^{(t)}(a|s)} \tag{47}$$

$$\ge \alpha \cdot \sum_{a \in I_s^{(t)-}} \pi^{(t)}(a|s) \quad (\because \sum_{a \in I_s^{(t)+}} \pi^{(t)}(a|s) \le 1) \tag{48}$$

$$\ge \alpha \cdot \frac{p\Delta}{\frac{2R}{1-\gamma}} \quad \text{(From inequality (45))} \tag{49}$$

for all $t > T_1$. $\square$

## A.5 PROOF OF LEMMA 4

For ease of exposition, we restate Lemma 4 as follows.

**Lemma.** $\sum_{a \in I_s^+ \cup I_s^-} \epsilon_s^{(t)} \pi^{(t)}(a|s) |A^{(t)}(s, a)| \to 0$ *as* $t \to \infty$.

*Proof.* Convergence of $V^{(t)}(s)$ implies that $V^{(t+1)}(s) - V^{(t)}(s) \to 0$, and by (4), the difference between the value functions at $t$ and $t+1$ can be written as

$$V^{(t+1)}(s) - V^{(t)}(s)$$
$$= \frac{1}{1-\gamma} \sum_{s' \in \mathcal{S}} d_s^{(t+1)}(s') \sum_{a \in \mathcal{A}} \pi^{(t+1)}(a|s') A^{(t)}(s', a). \tag{50}$$

Following the update rule given by our classification-based scheme, $\pi^{(t+1)}$ satisfies both Proposition 1 and Proposition 2. That is,

$$\sum_{a \in \mathcal{A}} \pi^{(t+1)}(a|s') A^{(t)}(s', a) \ge 0 \,\forall s' \in \mathcal{S}. \tag{51}$$

This implies that for convergence of $V^{(t)}(s)$, it is sufficient to show that

$$d_s^{(t+1)}(s') \sum_{a \in \mathcal{A}} \pi^{(t+1)}(a|s')A^{(t)}(s', a) \to 0, \text{ for all } s' \in \mathcal{S}. \tag{52}$$

Let us focus on $s' = s$. Simply considering the first term of the discounted state visitation frequency, we obtain

$$\sum_{t'=0}^{\infty} \gamma^{t'} \Pr(s_{t'} = s|s_0 = s, \pi^{(t+1)})$$

$$\geq \gamma^0 \Pr(s_0 = s|s_0 = s, \pi^{(t+1)}) = 1, \tag{53}$$

and thus

$$d_s^{(t+1)}(s) = (1 - \gamma) \sum_{t'=0}^{\infty} \gamma^{t'} \Pr(s_{t'} = s|s_0 = s, \pi^{(t+1)})$$

$$\geq 1 - \gamma. \tag{54}$$

From (52) and (54) we have

$$\sum_{a \in \mathcal{A}} \pi^{(t+1)}(a|s)A^{(t)}(s, a) \to 0. \tag{55}$$

Combining the above result and the fact that $\sum_{a \in A} \pi^{(t)}(a|s)A^{(t)}(s, a) = 0$, we have

$$\sum_{a \in \mathcal{A}} (\pi^{(t+1)}(a|s) - \pi^{(t)}(a|s))A^{(t)}(s, a) \to 0. \tag{56}$$

We split the summation over all actions into three parts, according to limits of the advantage function, i.e.,

$$\sum_{a \in \mathcal{A}} (\pi^{(t+1)}(a|s) - \pi^{(t)}(a|s))A^{(t)}(s, a)$$
$$= \sum_{a \in I_s^0} (\pi^{(t+1)}(a|s) - \pi^{(t)}(a|s))A^{(t)}(s, a)+$$
$$\sum_{a \in I_s^+} (\pi^{(t+1)}(a|s) - \pi^{(t)}(a|s))A^{(t)}(s, a)+$$
$$\sum_{a \in I_s^-} (\pi^{(t+1)}(a|s) - \pi^{(t)}(a|s))A^{(t)}(s, a) \tag{57}$$

Let $T = \max\{T_0, T_1\}$, where $T_0, T_1$ are the constants defined in Lemma 2. Then, for $a \in I_s^+ \cup I_s^-$, $A^{(t)}(s, a)$ is of the same sign as $A^{(\infty)}(s, a)$, for all $t > T$.

After moving the term of summation over $I_s^0$ in (57) to the left hand side, by our classification-based scheme, we have

$$\sum_{a \in \mathcal{A}} (\pi^{(t+1)}(a|s) - \pi^{(t)}(a|s))A^{(t)}(s, a) - \sum_{a \in I_s^0} (\pi^{(t+1)}(a|s) - \pi^{(t)}(a|s))A^{(t)}(s, a) \tag{58}$$

$$= \sum_{a \in I_s^+} (\pi^{(t+1)}(a|s) - \pi^{(t)}(a|s))A^{(t)}(s, a) + \sum_{a \in I_s^-} (\pi^{(t+1)}(a|s) - \pi^{(t)}(a|s))A^{(t)}(s, a) \tag{59}$$

$$= \sum_{a \in I_s^+} (\pi^{(t+1)}(a|s) - \pi^{(t)}(a|s))A^{(t)}(s, a) - \sum_{a \in I_s^-} (\pi^{(t+1)}(a|s) - \pi^{(t)}(a|s))|A^{(t)}(s, a)| \tag{60}$$

$$\geq \sum_{a \in I_s^+} \zeta\epsilon_s^{(t)}\pi^{(t)}(a|s)A^{(t)}(s, a) + \sum_{a \in I_s^-} \zeta\epsilon_s^{(t)}\pi^{(t)}(a|s)|A^{(t)}(s, a)| \tag{61}$$

$$= \sum_{a \in I_s^+ \cup I_s^-} \zeta\epsilon_s^{(t)}\pi^{(t)}(a|s)|A^{(t)}(s, a)| > 0, \tag{62}$$

for all $t > T$. Note that (61) follows from the same argument as (24)-(25). By the fact that

$$\sum_{a \in \mathcal{A}} (\pi^{(t+1)}(a|s) - \pi^{(t)}(a|s)) A^{(t)}(s, a) \to 0 \tag{63}$$

and

$$\sum_{a \in I_s^0} (\pi^{(t+1)}(a|s) - \pi^{(t)}(a|s)) A^{(t)}(s, a) \to 0, \tag{64}$$

we have (58) converges to zero. Finally, by the sandwich theorem, we have

$$\sum_{a \in I_s^+ \cup I_s^-} \epsilon_s^{(t)} \pi^{(t)}(a|s) |A^{(t)}(s, a)| \to 0. \tag{65}$$

$\square$

# B  PROOFS OF COROLLARIES 1 AND 2

## B.1  PROOF OF COROLLARY 1

From Theorem 1, we know that $I_s^+$ is an empty set, for all $s \in \mathcal{S}$. Therefore, the result of Lemma 4 becomes

$$\sum_{a \in I_s^-} \epsilon_s^{(t)} \pi^{(t)}(a|s) |A^{(t)}(s, a)| \to 0. \tag{66}$$

This implies that either $\epsilon_s^{(t)} \to 0$ or $\sum_{a \in I_s^-} \pi^{(t)}(a|s) \to 0$. By Lemma 2, $I_s^{(t)-} \supseteq I_s^-$, for all $t > T_1$. This implies that $\sum_{a \in I_s^{(t)-}} \pi^{(t)}(a|s) > \sum_{a \in I_s^-} \pi^{(t)}(a|s)$, for all $t > T_1$. Moreover, we have that $\epsilon_s^{(t)} \to 0$ implies $\sum_{a \in I_s^{(t)-}} \pi^{(t)}(a|s) \to 0$ and thereby implies $\sum_{a \in I_s^-} \pi^{(t)}(a|s) \to 0$, i.e., $\sum_{a \in I_s^-} \pi^{(t)}(a|s) \to 0$ is a necessary condition for $\epsilon_s^{(t)} \to 0$. Hence, we know

$$\sum_{a \in I_s^-} \pi^{(t)}(a|s) \to 0 \tag{67}$$

regardless of the behavior of $\epsilon_s^{(t)}$.

By combining $\sum_{a \in I_s^-} \pi^{(t)}(a|s) \to 0$ and the fact that $I_s^+$ is empty, we have

$$\sum_{a \in I_s^0} \pi^{(t)}(a|s) \to 1. \tag{68}$$

$\square$

## B.2  PROOF OF COROLLARY 2

Since there is only one action $a \in I_s^0$ and $I_s^+$ is empty, for all $a' \in \mathcal{A} \setminus \{a\}$, we have $a' \in I_s^-$. By Lemma 2, there exists $T_1$ such that for all $t > T_1$, $A^{(t)}(s, a') < 0$, for all $a' \in \mathcal{A} \setminus \{a\}$. This implies that $A^{(t)}(s, a) > 0$, for all $t > T_1$, i.e., for all $t > T_1$, we have $a \in I_s^{(t)+}$ and $a' \in I_s^{(t)-}$, for all $a' \in \mathcal{A} \setminus \{a\}$.

Then, by Corollary 1, as $t \to \infty$

$$\sum_{a \in I_s^{(t)+}} \pi^{(t)}(a|s) = \sum_{a \in I_s^0} \pi^{(t)}(a|s) \to 1 \tag{69}$$

$$\sum_{a \in I_s^{(t)-}} \pi^{(t)}(a|s) = \sum_{a \in I_s^-} \pi^{(t)}(a|s) \to 0. \tag{70}$$

Hence, by definition, $\epsilon_s^{(t)} = \alpha \cdot \min\{1, \frac{\sum_{a \in I_s^{(t)-}} \pi^{(t)}(a|s)}{\sum_{a \in I_s^{(t)+}} \pi^{(t)}(a|s)}\} \to 0$ as $t \to \infty$. $\square$

## C  PROOF OF THEOREM 2

For ease of exposition, we restate Theorem 2 as follows.

**Theorem.** *Theorem 1 is also satisfied for the following classifiers with corresponding margins with $0 < \alpha < 1$:*

*(i) For classifier $\log(\pi_\theta(a|s)) - \log(\pi(a|s))$, let*

$$\epsilon_s^\pi = \log(1 + \alpha \cdot \min\{1, \frac{\sum_{a \in I_s^{\pi-}} \pi(a|s)}{\sum_{a \in I_s^{\pi+}} \pi(a|s)}\}). \tag{71}$$

*(ii) For classifier $\sqrt{\rho_{s,a}(\theta)} - 1$, let*

$$\epsilon_s^\pi = \sqrt{1 + \alpha \cdot \min\{1, \frac{\sum_{a \in I_s^{\pi-}} \pi(a|s)}{\sum_{a \in I_s^{\pi+}} \pi(a|s)}\}} - 1. \tag{72}$$

*Proof.* The proof of Theorem 2 follows exactly the same procedure as that of Theorem 1. We sketch the proof for the two different classifiers: To begin with, we extend the supporting lemmas to HPO-AM with these two alternative classifiers:

- Notice that by using exactly the same argument as Lemma 5, we have that under these two classifiers, strict policy improvement $\pi_\theta > \pi^{(t)}$ is guaranteed if $\hat{L}_s(\theta_s) \leq (1 - \zeta)\delta_s^{(t)}\epsilon_s^{(t)}$ for all $s$, where $\zeta \in (0, 1)$.

- Regarding Lemma 6, the policy $\pi_\theta$ we construct in the proof of Lemma 6 also achieves zero loss for the alternative loss functions induced by the two alternative classifiers.

- Regarding Lemma 1, since both the logarithm function and the square root function are concave, and hinge loss is convex and non-increasing, their compositions preserve convexity (Boyd et al., 2004). Moreover, by the clipping property of hinge loss, we also know that in each iteration $t$, $\hat{L}_s(\theta_s)$ is Lipschitz continuous under the two classifiers $\log(\rho_{s,a}(\theta_s))$ and $\sqrt{(\rho_{s,a}(\theta_s))} - 1$ if the policy $\pi^{(t)}$ is strictly positive in all state-action pairs. Hence, $\hat{L}(\theta)$ also remains Lipschitz continuous. By using the same argument of convergence of EMDA as that in Lemma 1, it is easy to verify that $\pi^{(t+1)} > \pi^{(t)}$ and $\pi^{(t+1)}(a|s) > 0$ for all $(s, a)$ if EMDA takes diminishing and non-summable step sizes.

- Based on the extended version of Lemma 5 and Lemma 1 for these two alternative classifiers and Assumptions 3-4, we know that under the design of HPO-AM, strict policy improvement (i.e., $\pi^{(t+1)} > \pi^{(t)}$) and strict positivity of $\pi^{(t)}$ are satisfied in every iteration $t$. Therefore, the limits $V^{(\infty)}$, $Q^{(\infty)}$, and $A^{(\infty)}$ exist by monotone convergence theorem.

- Lemma 2 does not use the specific form of the classifier and hence still holds under the two alternative classifiers.

- For the two alternative classifiers, Lemma 3 can be readily extended by adapting the argument in (46)-(49) to the $\epsilon_s^\pi$ defined in (19) and (20). Similarly, Lemma 4 can be extended by adapting the inequalities in (61)-(62) to the corresponding $\epsilon_s^\pi$ defined in (19) and (20).

Since all the lemmas hold for these alternative classifiers and their corresponding $\epsilon_s^\pi$, we can complete the proof by showing a contradiction similar to that in Theorem 1.

$\square$

## D  UNPROVEN CLASSIFIERS

In Remark 6, we identify two other classifiers, $\pi_\theta(a|s) - \pi(a|s)$ and $(\rho_{s,a}(\theta))^2 - 1$, which we cannot directly apply similar proof as Theorem 1.

First, an intuitive reason for the classifier $\pi_\theta(a|s) - \pi(a|s)$ is explained as follows. To ensure the existence of improved policy, we must set $\epsilon_s = \min_{a \in \mathcal{A}}\{\pi(a|s)\} \cdot \alpha \cdot \min\{1, \frac{\sum_{a \in I_s^{\pi-}} \pi(a|s)}{\sum_{a \in I_s^{\pi+}} \pi(a|s)}\}, 0 < \alpha < 1$. Note that there is an additional term in front, compared to the margin we use in Theorem 1 before. Due to multiplying this additional term, the margin is bounded by the minimum probability of action. We cannot use Lemma 3 to give a lower bound for $\epsilon_s^{(t)}$, since $\min_{a \in A}\{\pi^{(t)}(a|s)\}$ will probably approach zero as $t \to \infty$. To be more specific, it makes the probability of all the other actions increase/decrease slowly, when there exists an action with extremely small probability. Without Lemma 3, we are not able to prove the global convergence by contradiction.

Second, for the classifier $(\rho_{s,a}(\theta))^2 - 1$, although we can prove the existence of improved policy by defining the margin as $\epsilon_s = (1 + \alpha \cdot \min\{1, \frac{\sum_{a \in I_s^{\pi-}} \pi(a|s)}{\sum_{a \in I_s^{\pi+}} \pi(a|s)}\})^2 - 1$, our loss function is no longer convex. Without convexity, the EMDA method is not guaranteed to converge to the optimal value. We leave it as an interesting open question of whether or not this problem can be solved by using a non-convex optimization method.

# E    ABLATION STUDY OF HPO-AM VIA EXPERIMENTS IN A TABULAR ENVIRONMENT

## E.1    GRIDWORLD ENVIRONMENT

For a principled discussion, we use a $4 \times 4$ gridworld similar to that in (Sutton, 1988, Example 4.1), where an optimal policy and all the value functions can be analytically obtained. Regarding the gridworld environment, there are four available actions (up, down, right, left) and a terminal state located at the top-left corner. At each non-terminal state, the reward is either 1.0 or -1.2 with equal probability, for all the actions. All the state transitions are deterministic.

## E.2    EXPERIMENTAL SETTINGS

In the gridworld environment, all the policies in our experiments are trained for 40000 episodes under 10 fixed random seeds. All the source code is in `supplementary.zip` of the supplementary materials. Our code of the HPO algorithms is extended from an open-source PyTorch example[3], and the gridworld environment is modified from an open-source project, gym-gridworld[4].

## E.3    NEURAL NETWORK MODEL

Our neural network model is composed of three layers: two fully connected layers and one output layer, which includes two output heads for the value and the action. The two fully connected layers with 32 hidden units are shared for the output heads. The learning rate and the discount factor ($\gamma$) are set to be 0.001 and 0.99, respectively. The input of the model is a one-hot vector to represent the grid position.

## E.4    ABLATIONS FOR THE WEIGHT AND MARGIN

Recall that the loss function of HPO in (12) involves three critical components: a classifier, a weight, and a margin. Regarding the classifier, we evaluate five variants of HPO-AM with the classifiers , including: (i) $\rho_{s,a}(\theta) - 1$, (the standard HPO-AM); (ii) $\log(\pi_\theta(a|s)) - \log(\pi(a|s))$ (HPO-AM-log); (iii) $\sqrt{\rho_{s,a}(\theta)} - 1$ (HPO-AM-root); (iv) $\pi_\theta(a|s) - \pi(a|s)$ (HPO-AM-sub); (v) $(\rho_{s,a}(\theta))^2 - 1$ (HPO-AM-square). Regarding the weight and the margin, we consider four different scenarios, including: (i) adaptive $\epsilon_s$ and $|\hat{A}^{(t)}(s_k, a_k)|$ as the weight (denoted as WAE); (ii) adaptive $\epsilon_s$ and a constant weight of 1 (denoted as AE); (iii) constant $\epsilon_s = 0.1$ and $|\hat{A}^{(t)}(s_k, a_k)|$ as the weight (denoted as WCE); (iv) constant $\epsilon_s = 0.1$ and a constant weight of 1 (denoted as CE). For the constant margin, we follow the popular setting of PPO-clip to set $\epsilon = 0.1$. Those methods are implemented with a

---

[3]`https://github.com/pytorch/examples/blob/master/reinforcement_learning/actor_critic.py`

[4]`https://github.com/podondra/gym-gridworlds`

tabular policy or a neural network policy. Under all the algorithms, the policy is trained for 40000 episodes under 10 random seeds.

### E.5 TRAINING CURVES

We show the $L_1$ norm between HPO-AM policy with the optimal policy in Figure 2 and Figure 3. First, we evaluate the five variants of HPO-AM under tabular parameterization with true advantage values in Figure 2. This experiment is meant to verify the convergence behavior of HPO-AM.

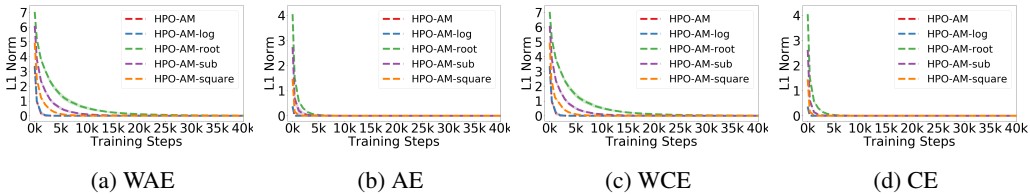

|     (a) WAE     |     (b) AE     |     (c) WCE     |     (d) CE     |

Figure 2: Experimental results of tabular policies in the gridworld environments.

We further implement the five variants of HPO-AM using neural networks with estimated advantage values. Figure 3 reveals the mean value and the standard deviation of $L_1$ norm. Most of the five algorithms can converge within a reasonable number of training episodes, while some of HPO-AM-sub with WAE and WCE settings cannot converge.

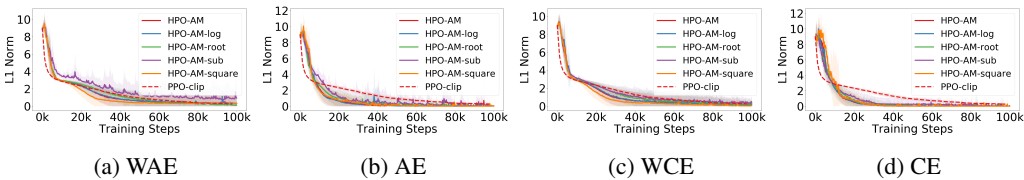

|     (a) WAE     |     (b) AE     |     (c) WCE     |     (d) CE     |

Figure 3: Experimental results of neural network policies in the gridworld environments.

### E.6 DISCUSSIONS

We measure the $L_1$ norm between the policy learned under each variant of HPO-AM with the optimal policy as the performance criterion. First, we evaluate the five variants of HPO-AM under tabular parameterization with true advantage values. This experiment is meant to verify the convergence behavior of HPO-AM. As expected, HPO-AM, HPO-AM-log and HPO-AM-root indeed converge to the optimal policy (in a small number of training episodes). Interestingly, HPO-AM-sub and HPO-AM-square, which are based on the two weak classifiers without theoretical guarantees as discussed in the previous section, still converges. These results serve as empirical insights into the convergence analysis for these two classifiers. We further implement the five variants of HPO-AM using neural networks with estimated advantage values. Most of the five algorithms can converge within a reasonable number of training episodes. We also observe that the HPO-AM algorithms appear to converge slightly faster than PPO-clip. This manifests the potential of the proposed HPO framework in opening up a new class of promising RL algorithms.

## F DETAILED CONFIGURATIONS OF THE EXPERIMENTS IN MINATAR

### F.1 EXPERIMENTAL SETTINGS

For the experiments in the MinAtar environments (Young & Tian, 2019), we implement HPO-AM and its variants with WAE setting on top of the open-source RL Baselines 3 Zoo framework (Raffin, 2020). Recall the classifier is a critical component for the loss function of HPO in (12), we evaluate five variants of HPO-AM with the classifiers, including: (i) $\rho_{s,a}(\theta) - 1$, (the standard

HPO-AM); (ii) $\log(\pi_\theta(a|s)) - \log(\pi(a|s))$ (HPO-AM-log); (iii) $\sqrt{\rho_{s,a}(\theta)} - 1$ (HPO-AM-root); (iv) $\pi_\theta(a|s) - \pi(a|s)$ (HPO-AM-sub); (v) $(\rho_{s,a}(\theta))^2 - 1$ (HPO-AM-square). In MinAtar, the observation is encoded by $10 \times 10 \times n$ state representations, where each of the $n$ channels corresponds to a game-specific object. For example, in Breakout, the game-specific objects are ball, paddle and brick. We compare HPO-AM variants with PPO-clip, A2C, and Rainbow in Asterix, Breakout and Freeway with 5 million training steps among 5 random seeds. The source code is in `supplementary.zip` of the supplementary materials.

### F.2 NEURAL NETWORK MODEL

The network model of HPO-AM is composed of two parts: a shared feature extractor and an output layer (including two sets of output heads, one for the Q-value and the other for the policy). For the MinAtar experiments, the feature extractor consists of one 2D convolution layer (with $3 \times 3$ kernel and 16 output features) and one fully connected layer with 128 hidden units. Regarding A2C and PPO-clip, we set the entropy coefficient to 0 to disable the entropy regularization for a fair comparison and follow the configuration suggested by RL Baseline3 Zoo[5] for the rest of the hyperparameters. Table 1 summarize the hyperparameters for MinAtar and all HPO-AM variants use the same setting. For each MiniAtar game, we run the experiments under 5 random seeds with 5,000,000 training steps for all methods.

Table 1: Parameters for the MinAtar experiments (lin_2.5e-4 means that the learning rate decays linearly from $2.5 \times 10^{-4}$ to 0).

| Hyperparameters | HPO-AM | PPO-clip | A2C |
|---|---|---|---|
| n_envs | 8 | 8 | 16 |
| n_steps | 64 | 128 | 5 |
| batch_size | 256 | 512 | 80 |
| n_epochs | 10 | 4 | - |
| learning_rate | lin_2.5e-4 | lin_2.5e-4 | 7e-4 |
| vf_coef | 0.8 | 0.5 | 0.5 |
| $\alpha$ | 0.5 | - | - |

## G COMPARISON OF THE CLIPPED OBJECTIVE IN PPO-CLIP AND THE HINGE LOSS OBJECTIVE OF HPO

Recall that the original objective of PPO-clip is

$$L^{\text{clip}}(\theta) = \mathbb{E}_{s \sim d^\pi_{\mu_0}, a \sim \pi(\cdot|s)} \big[ \min\{\rho_{s,a}(\theta) A^\pi(s,a), \text{clip}(\rho_{s,a}(\theta), 1-\epsilon, 1+\epsilon) A^\pi(s,a)\}\big], \quad (73)$$

where $\rho_{s,a}(\theta) = \frac{\pi_\theta(a|s)}{\pi(a|s)}$. In practice, $L^{\text{clip}}(\theta)$ is approximated by the sample average as

$$L^{\text{clip}}(\theta) \approx \hat{L}^{\text{clip}}(\theta) = \frac{1}{|D_\pi|} \sum_{(s,a) \in D_\pi} \min\{\rho_{s,a}(\theta) A^\pi(s,a), \text{clip}(\rho_{s,a}(\theta), 1-\epsilon, 1+\epsilon) A^\pi(s,a)\}$$

$$(74)$$

$$= \frac{1}{|D_\pi|} \sum_{(s,a) \in D_\pi} |A^\pi(s,a)| \cdot \underbrace{\min\{\rho_{s,a}(\theta) \text{sign}(A^\pi(s,a)), \text{clip}(\rho_{s,a}(\theta), 1-\epsilon, 1+\epsilon) \text{sign}(A^\pi(s,a))\}}_{=:W^{\text{clip}}_{s,a}(\theta)}.$$

$$(75)$$

---

[5] `https://github.com/DLR-RM/rl-baselines3-zoo/blob/master/benchmark.md`

Note that $W_{s,a}^{\text{clip}}(\theta)$ can be further written as

$$
W_{s,a}^{\text{clip}}(\theta) = \begin{cases} 1 + \epsilon & , \text{if } A^\pi(s,a) > 0 \text{ and } \rho_{s,a}(\theta) \geq 1 + \epsilon \\ \rho_{s,a}(\theta) & , \text{if } A^\pi(s,a) > 0 \text{ and } \rho_{s,a}(\theta) < 1 + \epsilon \\ -\rho_{s,a}(\theta) & , \text{if } A^\pi(s,a) < 0 \text{ and } \rho_{s,a}(\theta) > 1 - \epsilon \\ -(1 - \epsilon) & , \text{if } A^\pi(s,a) < 0 \text{ and } \rho_{s,a}(\theta) \leq 1 - \epsilon \\ 0 & , \text{otherwise} \end{cases}
$$

Recall from (11) that the loss function of HPO takes the form as

$$
L(\theta) \approx \hat{L}(\theta) = \frac{1}{|D_\pi|} \sum_{(s,a) \in D_\pi} |A^\pi(s,a)| \cdot \underbrace{\max\left\{0, \epsilon - (\rho_{s,a}(\theta) - 1)\,\text{sign}(A^\pi(s,a))\right\}}_{=:W_{s,a}(\theta)}. \tag{76}
$$

Similarly, $W_{s,a}(\theta)$ can be further written as

$$
W_{s,a}(\theta) = \begin{cases} 0 & , \text{if } A^\pi(s,a) > 0 \text{ and } \rho_{s,a}(\theta) \geq 1 + \epsilon \\ -\rho_{s,a}(\theta) + (1 + \epsilon) & , \text{if } A^\pi(s,a) > 0 \text{ and } \rho_{s,a}(\theta) < 1 + \epsilon \\ \rho_{s,a}(\theta) - (1 - \epsilon) & , \text{if } A^\pi(s,a) < 0 \text{ and } \rho_{s,a}(\theta) > 1 - \epsilon \\ 0 & , \text{if } A^\pi(s,a) < 0 \text{ and } \rho_{s,a}(\theta) \leq 1 - \epsilon \\ \epsilon & , \text{otherwise} \end{cases}
$$

Therefore, it is easy to verify that $\hat{L}^{\text{clip}}(\theta)$ and $-\hat{L}(\theta)$ only differ by a constant with respect to $\theta$. This also implies that $\nabla_\theta \hat{L}^{\text{clip}}(\theta) = -\nabla_\theta \hat{L}(\theta)$.

