# OpenReview forum: "Hinge Policy Optimization: Rethinking Policy Improvement and Reinterpreting PPO"
_ICLR.cc/2022/Conference — ICLR 2022 Submitted_

### Official Review · Reviewer_PyUp · 2021-11-01

**Correctness:** 4
**Technical Novelty And Significance:** 3
**Empirical Novelty And Significance:** 2
**Recommendation:** 8
**Confidence:** 2

**Main Review:**

I think this is a very interesting paper that improves a commonly used algorithm and provides both theoretical and empirical evidence that support using this modified algorithm.
The paper is very well-written both in terms of how the main parts of the paper are easy to follow, and the appendix section is very detailed.

Besides minor corrections and clarifications below, the only "weakness" is that PPO is a heavily used algorithm and the empirical results here are not convincing enough (only a part of the mini atar benchmark and a really small toy problem). However, in my opinion, the theoretical arguments provide enough value for the RL community so even though a more through survey would be valuable for practitioners, the current contents of this work are already enough for a good paper.

Minor issues:
1. eq 8 = eq 9?
2. In "Connecting PPO-clip and Hinge Loss" section: In practical -> In practice (?)
3. Regarding:
"and |Aπ (s, a)| serves as the weight of the sample."
I think you meant that A could be interpreted as the weight, but on initial read I thought you added an additional multiplicative term of A. Maybe rephrase to clarify it is not an additional modification, but an interpretation of an existing element.
4. Eq 13: what happens if all the advantages for a given state are 0? (I suspect that the denominator should be greater or equal to 0, since at least one action is non-negative)
5. Could you clarify the following statement in assumption 3:
" Strict positivity of the initial state distribution µ0 and the initial policy π (0) is a necessary condition for this assumption"
6. In remark 3:
"we leverage Assumption 3 in order to rigorously" - assumption 4? assumptions 3 and 4? assumption 3 alone seems out of context. And you need both the data assumption and the advantage sign assumption to prove your claims, right?
7. Eq 15: if I understand correctly, this expression is for the true advantage in the limit, which should be the empty set if the algorithm converges. You don't mention this expression in the remarks directly below. I found it confusing that you don't mention it at all, you probably meant to present it for completeness. It might be helpful to hint on its role in the text as well.

**Summary Of The Paper:**

This paper suggests a generalization to the commonly used PPO(-clip) algorithm, by analyzing the loss function from a state-wise perspective and generalizing the policy-ratio clip objective into a hinge loss formulation. The main contributions of this paper are:
(1) They introduce a family of algorithm that includes PPO-clip
(2) Show convergence results for these algorithms (under less strict assumptions than previous works)
(3) They suggest an instance from the suggested family of algorithms  - HPO-AM
(4) They provide some empirical evidence (specifically on some of the mini atar games) showing this algorithm is competitive with other commonly used algorithms.

**Summary Of The Review:**

I found this paper to be valuable to the deep-RL community as it presents an interesting generalization to a commonly used algorithm with theoretical justifications.

---

> ### Author Response · Authors · 2021-11-21
> **Response to Reviewer PyUp**
>
> We greatly appreciate the reviewer for the positive feedback and the detailed suggestions for improving our paper. We provide our point-by-point response as follows:
>
>
> #### (D1) "Regarding: 'and $\lvert A^{\pi}(s,a)\rvert$ serves as the weight of the sample.' I think you meant that A could be interpreted as the weight, but on initial read I thought you added an additional multiplicative term of A. Maybe rephrase to clarify it is not an additional modification, but an interpretation of an existing element."
>
> Thank you for the suggestion. We have rephrased this sentence in the updated manuscript as “$\lvert A^{\pi}(s,a)\rvert$ can be interpreted as the weight or cost associated with each sample in large-margin classification.”
>
>
> #### (D2) "In Eq(13), what happens if all the advantages for a given state are 0?"
>
> In this case, all the actions are equally good for this specific state $s$, and by the performance difference lemma [Kakade and Langford 2002], changing the action distribution at this state $s$ does not change the value at any state. Based on this, we could simply let $\epsilon_s^\pi=0$.
>
> #### (D3) Clarify the following statement in Assumption 3: "Strict positivity of the initial state distribution $\mu_0$ and the initial policy $\pi$ is a necessary condition for this assumption."
>
> Since the standard PPO-clip is on-policy, we consider HPO-AM and its variants to also be on-policy algorithms in order to better connect HPO-AM and PPO-clip. Therefore, the strict positivity conditions are necessary to ensure that the sampled training data set during each iteration of HPO-AM contains all possible state-action pairs.
>
> #### (D4) "Eq 15: if I understand correctly, this expression is for the true advantage in the limit, which should be the empty set if the algorithm converges. You don't mention this expression in the remarks directly below. I found it confusing that you don't mention it at all, you probably meant to present it for completeness. It might be helpful to hint on its role in the text as well."
>
> Thank you for pointing this out. We introduce the notation $I_s^+$ for the completeness of the discussion in Corollaries 1-2 and for the proof of Theorem 1. In the updated manuscript, we have added one sentence after Eqs (15)-(17) to describe its role.
>
> We have also fixed the typos mentioned in the review and updated the manuscript.

---

### Official Review · Reviewer_FV8P · 2021-11-01

**Correctness:** 3
**Technical Novelty And Significance:** 3
**Empirical Novelty And Significance:** 2
**Recommendation:** 6
**Confidence:** 4

**Main Review:**

Please list both the strengths and weaknesses of the paper. When discussing weaknesses, please provide concrete, actionable feedback on the paper.

**Originality:** To the best of my knowledge, this is the first formalization which allows a theoretical analysis of PPO-clip, albeit, as highlighted in the paper, not the first formalization of reinforcement learning as a classification problem.

**Significance:** Since PPO-clip is one of the most commonly used policy gradient algorithms, a theoretical analysis of the approach is a relevant contribution to the reinforcement learning community.

**Rigour:** The paper is built upon common tools and I could not find any factual errors in the proofs.

**Strengths**

- The paper provides a nice bridge between the existing theoretical tools for the analysis of policy gradient algorithms and the popular PPO-clip algorithm.
- The presentation of the theoretical results is quite clear.

**Major Concerns**

- I do not fully understand the justification behind the use of the specific EMDA approach as a subroutine in the algorithm. Is this essential for proving convergence of the resulting algorithm? Or would an alternative, perhaps simpler, procedure yield convergence as well albeit with a different proof scheme?
- The fact that the experimental results are relegated in Appendix is quite disappointing. Albeit they are in small experimental settings and the results are not stellar, they provide a nice additional perspective on the proposed framework. In my opinion, it could take the space now reserved to the proof of Theorem 1, which generally follows (Agarwal et al. 2020).
- The simple tabular policy representation used in the theoretical analysis burns some of the bridges between theory and practice that the paper is trying to build, since PPO-clip is traditionally associated to neural function approximators. What worries me the most is how the optimality in Definition 1-2 transfers to the setting with function approximation, in which this kind of sorting among policies is often unfeasible.

**Minor Concerns**

- Before Equation (8), it is written that the hinge loss and the PPO-clip gradients differ by a constant, which is not correct, given that the advantage depends on states and actions.
- Why is Equation (8)-(9) repeated?

**Summary Of The Paper:**

The paper proposes hinge policy optimization, a new theoretical framework for interpreting policy gradient algorithms as classification problems to be solved with a hinge loss. In this perspective, the sign of the advantage function becomes the label, and the difference in action probabilities between policy after and before an update becomes the classifier's output. The paper shows the equivalence between such a formulation and the popular PPO-clip objective and provides global converge guarantees on the blueprint of (Agarwal, 2020). It also proposes a range of policy optimization algorithms, depending on the details of the classification algorithm that is used, and empirically evaluates some of them.

**Summary Of The Review:**

The paper mostly achieves its goal, which is to provide a theoretical justification for PPO-clip. I currently lean towards acceptance, although the paper suffers from some presentation issues and imperfect justification of some algorithmic/theoretical design choices.

---

> ### Author Response · Authors · 2021-11-21
> **Response to Reviewer FV8P (1/2)**
>
> We sincerely thank the reviewer for the overall positive feedback and helpful suggestions. We provide our point-by-point response as follows.
>
> #### (C1) “Is EMDA essential for proving convergence of the resulting algorithm? Or would an alternative, perhaps simpler, procedure yield convergence as well albeit with a different proof scheme?”
>
> We leverage EMDA to minimize the loss function $\hat{L}^{(t)}(\theta)$ of HPO-AM in order to ensure that $\hat{L}^{(t)}(\theta)$ is Lipschitz continuous throughout training (cf. Eqs (35)-(39) in Appendix A). Specifically, as the loss function of HPO (and PPO-clip as well) involves the ratio $\pi_{\theta}(a|s)/\pi(a|s)$ between the new and the old policies, it is required to ensure the strict positivity of the policy, i.e., $\pi(a|s)>0$ for all $(s,a)$ (cf. Lemma 1 in the updated manuscript). On top of that, as the standard PPO-clip is an on-policy method, strict positivity of the policy is needed to avoid a lack of exploration.
>
> On the other hand, it appears possible to apply other constrained optimization techniques, such as the projected subgradient method, to replace the EMDA subroutine in Line 6 of Algorithm 1. In that case, it would also be required to show the strict positivity of the policy under such a constrained optimization scheme.
>
> #### (C2) Regarding the experimental results in the appendix
>
> Thank you for the suggestion. As one major contribution of this paper is to propose HPO and establish the global convergence of HPO, we expect that keeping the main proof of Theorem 1 and the statement of the supporting lemmas (with details moved to appendix) in the main text would greatly help the readers understand the essence of HPO. On top of that, we also agree with the reviewer that moving some part of the experimental results could further strengthen this work. As a result, in the updated manuscript, we have managed to accommodate the MinAtar results in the main text by making the proof of Theorem 1 and the part of supporting lemmas more compact.
>
> #### (C3) Explain how the optimality in Definitions 1-2 transfers to the setting with function approximation.
>
> Definitions 1-2 can be extended to the problem settings where the state space is countably infinite or continuous (the settings where function approximation is needed).
> For example:
> - In [Feinberg 2010], Definitions 1-2 have been adopted for MDPs with a countably infinite state space. It has been shown that if the state space is countable and the action space is compact and Borel (along with some mild conditions of the reward function and the transition probabilities), then an optimal policy in the sense of Definition 2 indeed exists (Corollary 8 of [Feinberg 2010]).
> - Moreover, the optimality in Definition 2 has been considered for MDPs with a continuous state space [Feinberg 2010]. If the state space is Borel and the action space is compact (along with some mild conditions of the reward function and the transition probabilities), then an optimal policy in the sense of Definition 2 exists (Theorem 16 of [Feinberg 2010]). More details can also be found in Chapter 9 of [Bertsekas & Shreve 1996].
>
> As the first step toward understanding the theoretical foundation of PPO-clip, this paper establishes the global convergence of PPO-clip for MDPs with finite state and action spaces. We expect that this could help pave the way toward better understanding PPO-clip in more general settings.
>
> [Bertsekas & Shreve 1996] Dimitri P. Bertsekas and Steven E. Shreve, “Stochastic Optimal Control: The Discrete Time Case,” 1996.
>
> [Feinberg 2010] Eugene A. Feinberg, "Total Expected Discounted Reward MDPs: Existence of Optimal Policies." Wiley Encyclopedia of Operations Research and Management Science, 2010.

---

> ### Author Response · Authors · 2021-11-21
> **Response to Reviewer FV8P (2/2)**
>
> #### (C4) Explain why the hinge loss and the negative of PPO-clip objective differ by a constant.
>
> The main idea is to clearly discuss the values of the two objectives under the different signs of $A^{\pi}(s,a)$ and $\rho_{s,a}(\theta)-1$.
> Recall that the original objective of PPO-clip is
> $$L^{\text{clip}}(\theta) =E_{s \sim d_{\mu_0}^{\pi},a \sim \pi(\cdot|s)} \big[\min\{\rho_{s,a}(\theta)A^{\pi}(s,a), \text{clip}(\rho_{s,a}(\theta),1-\epsilon,1+\epsilon)A^{\pi}(s,a)\}\big],$$
> where $\rho_{s,a}(\theta)=\frac{\pi_{\theta}(a\rvert s)}{\pi(a\rvert s)}$.
>
> In practice, $L^{\text{clip}}(\theta)$ is approximated by the sample average as
> $$
> \begin{aligned}
>     L^{\text{clip}}(\theta) \approx \hat{L}^{\text{clip}}(\theta)&=\frac{1}{\lvert D_\pi\rvert} \sum_{(s,a)\in D_{\pi}}\min\{\rho_{s,a}(\theta)A^{\pi}(s,a), \text{clip}(\rho_{s,a}(\theta),1-\epsilon,1+\epsilon)A^{\pi}(s,a)\} \\\\
>     &=\frac{1}{\lvert D_\pi\rvert} \sum_{(s,a)\in D_{\pi}}\lvert A^{\pi}(s,a)\rvert \cdot \underbrace{\min\lbrace\rho_{s,a}(\theta)\text{sign}(A^{\pi}(s,a)), \text{clip}(\rho_{s,a}(\theta),1-\epsilon,1+\epsilon)\text{sign}(A^{\pi}(s,a))\rbrace}_{W^{\text{clip}}(s,a;\theta)}.
> \end{aligned}
> $$
>
> Note that ${W}^{\text{clip}}(s,a;\theta)$ can be further written as
> $$
> W^{\text{clip}}(s,a;\theta)=
> \begin{cases}
>         1+\epsilon&, \text{if } A^{\pi}(s,a)>0 \text{ and } \rho_{s,a}(\theta)\geq 1+\epsilon  \\\\
>         \rho_{s,a}(\theta)&, \text{if } A^{\pi}(s,a)>0 \text{ and } \rho_{s,a}(\theta)<1+\epsilon\\\\
>         -\rho_{s,a}(\theta)&, \text{if } A^{\pi}(s,a)<0 \text{ and } \rho_{s,a}(\theta)>1-\epsilon\\\\
>         -(1-\epsilon)&, \text{if } A^{\pi}(s,a)<0 \text{ and } \rho_{s,a}(\theta)\leq 1-\epsilon\\\\
>         0&, \text{otherwise}
>     \end{cases}
> $$
>
> Recall from Eq (11) that the loss function of HPO takes the form as
> $$
> \begin{aligned}
>     L(\theta) &\approx \hat{L}(\theta)=\frac{1}{\lvert D_\pi\rvert} \sum_{(s,a)\in D_{\pi}}\lvert A^{\pi}(s,a)\rvert \cdot\underbrace{\max\big\lbrace 0,\epsilon-(\rho_{s,a}(\theta)-1)\text{sign}(A^{\pi}(s,a))\big\rbrace}_{=:W(s,a;\theta)}.
> \end{aligned}
> $$
>
> Similarly, $W(s,a;\theta)$ can be further written as
> $$
> W(s,a;\theta)=
> \begin{cases}
>         0&, \text{if } A^{\pi}(s,a)>0 \text{ and } \rho_{s,a}(\theta)\geq 1+\epsilon  \\\\
>         -\rho_{s,a}(\theta)+(1+\epsilon)&, \text{if } A^{\pi}(s,a)>0 \text{ and } \rho_{s,a}(\theta)<1+\epsilon\\\\
>         \rho_{s,a}(\theta)-(1-\epsilon)&, \text{if } A^{\pi}(s,a)<0 \text{ and } \rho_{s,a}(\theta)>1-\epsilon\\\\
>         0&, \text{if } A^{\pi}(s,a)<0 \text{ and } \rho_{s,a}(\theta)\leq 1-\epsilon\\\\
>         \epsilon&, \text{otherwise}
>     \end{cases}
> $$
>
> Therefore, it is easy to verify that $\hat{L}^{\text{clip}}(\theta)$ and $-\hat{L}(\theta)$ only differ by a constant with respect to $\theta$. This also implies that $\nabla_{\theta}\hat{L}^{\text{clip}}(\theta)= -\nabla_{\theta}\hat{L}(\theta)$.
>
> To clearly explain this, we have included the detailed comparison of these two objectives in Appendix G and added one sentence pointing to this in Page 4.
>
> #### (C5) Equations (8) and (9) are repeated.
>
> Thank you for catching this. We have made the changes in the updated manuscript.

---

### Official Review · Reviewer_vtrj · 2021-11-02

**Correctness:** 3
**Technical Novelty And Significance:** 3
**Empirical Novelty And Significance:** 2
**Recommendation:** 5
**Confidence:** 4

**Main Review:**

Pros:

-  the paper gives interesting theoretical insight linking PPO-clip with a large margin classification and theoretical analysis of a new group of reinforcement learning as classification methods.

Cons:

(1) while the reviewer appreciates the paper is mostly theoretical, it might be a good idea to bring some experimental analysis into the main text to show the performance of the method on a range of benchmark problems, especially given that the experimental analysis is already presented in the appendix. While the proofs are important part of the work, are they more essential for the narrative than the experiments? There is no right or wrong answer on this question, this question is more about gaining insight into why it has been done this way.

(2)restrictive assumptions of the proposed method: “Assumption 4. We assume that at each iteration t, for each state-action pair, the sign of the estimated advantage is the same as that of the true advantage. “;   "Assumption 2. Tabular policies” . Although it makes sense to  use assumption 4 for the sake of theoretical analysis, it would be good to have some evidence whether it actually could hold in any experimental scenarios. The reviewer wonders whether it is possible to show some experimental analysis in the appendix showing whether the assumption 4 holds during the training (and how it is reflected in training dynamics)?

Other comments:

(1) Page 5: Entropic mirror descent: could the authors give some references?

(2) In the experimental section of the appendix, it would be useful to see the grid world tasks compared with a well-known baseline the same way as the MinAtar problems are. It would be also useful to see more environments to show the advantages and limitations of the proposed idea on more scenarios.

**Summary Of The Paper:**

The paper describes a new family of methods, called Hinge Policy Optimisation (HPO), generalising over proximal policy optimisation algorithm with a clipped surrogate objective (PPO-clip).

**Summary Of The Review:**

Pros: novel idea and interesting analysis of reinforcement learning as classification
Cons: restrictive assumptions of the theoretical analysis; the experimental analysis could be given more space given that it actually exists in the appendix.
Starting from 5, but happy to update the scores to acceptance if there's a convincing answer on the cons during the rebuttal.

---

> ### Author Response · Authors · 2021-11-21
> **Response to Reviewer vtrj**
>
> We greatly appreciate the reviewer’s constructive feedback for improving our paper. We provide our point-by-point response as follows.
>
> #### (B1) “While the reviewer appreciates the paper is mostly theoretical, it might be a good idea to bring some experimental analysis into the main text to show the performance of the method on a range of benchmark problems, especially given that the experimental analysis is already presented in the appendix.”
>
> Thank you for the suggestion. As the major contribution of this paper is to propose HPO and establish the global convergence of HPO, we expect that keeping the main proof of Theorem 1 and the statement of the supporting lemmas in the main text (with the proof details in the appendix) would greatly help the readers understand the essence of HPO. On top of that, we also agree with the reviewer that moving some part of the experimental results could further strengthen this work. As a result, in the updated manuscript, we have managed to accommodate the MinAtar results in the main text by making the proof of Theorem 1 and the part of supporting lemmas more compact.
>
> #### (B2) “Although it makes sense to use assumption 4 for the sake of theoretical analysis, it would be good to have some evidence whether it actually could hold in any experimental scenarios.”
>
> To further justify Assumption 4, we rerun the gridworld experiments of Figure 2 in the appendix and record the estimated advantage function (learned by maintaining a Q-network) as well as the true advantage function. Notably, despite that the gridworld is of size 4x4, the fact that the random reward obtained at each step is either 1.0 or -1.2 with equal probability makes it non-trivial to estimate the signs of the advantage values. The following table shows the percentage of the state-action pairs at which the sign of the estimated advantage is correct, in the earlier and the later stages of the training process:
> (The results below are the average over 10 random seeds)
>
> | Algorithm| HPO-AM | HPO-AM-log | HPO-AM-root | HPO-AM-sub | HPO-AM-square |
> |-|---|---|---|---|---|
> | 20k steps | 89% | 98% | 89% | 90% | 95% |
> | 50k steps | 96% | 99% | 91% | 94% | 96% |
> | 100k steps | 97% | 100% | 88% | 93% | 99% |
>
> The above shows that Assumption 4 could be met well in the gridworld with random rewards.
>
> #### (B3) Provide a reference on entropic mirror descent
>
> The entropic mirror descent method has been analyzed and well discussed by [Beck & Teboulle, 2003], which is cited in Appendix A.3 for the proof of Lemma 3 in our original manuscript. To make this more clear, we have added this reference in Page 5 of the main text.
>
> [Beck & Teboulle, 2003] Amir Beck and Marc Teboulle, “Mirror descent and nonlinear projected subgradient methods for convex optimization,” Operations Research Letters, 2003.
>
> #### (B4) It would be useful to see the grid world tasks compared with a well-known baseline the same way as the MinAtar problems are.
>
> Thank you for the suggestion. To make the gridworld experiments more thorough, we further test PPO-clip (under the same 10 random seeds as HPO-AM) and add the experimental results to Figure 3 in the updated manuscript. We observe that the HPO-AM algorithms appear to converge slightly faster than PPO-clip.

---

> > ### Comment · Reviewer_vtrj · 2021-11-23
> > **Follow-up question**
> >
> > First of all, I would like to confirm that the rebuttal looks like a step in the right direction towards aleviating the proposed limitations. I think the structure has improved now, and I am happy to see some experimental results in the main text. There is, however, a follow up question on Assumption 4. This is to understand that the authors can show how it is possible to bridge the gap between the actual noisy labels of the estimated advantage signs and the proposed method assuming the signs are correct.
> >
> > It is possible to think that as the experiment shows roughly 10% of the labels are noisy, these might be exactly the samples which hinder the convergence. Therefore, one might think  of using the large margin classifiers which take into consideration the noise. For example, Natarajan et al (2013) Learning with Noisy Labels ( https://proceedings.neurips.cc/paper/2013/file/3871bd64012152bfb53fdf04b401193f-Paper.pdf ) propose a surrogate loss function which takes into account noise tolerance. I wonder if the proposed analysis is compatible with such noise functions and whether it may be the answer which bridges this gap. I accept that it may be still impossible, but explaining how the options of handling noise  in large margin classifiers could help increase the robustness towards this noise and therefore achieve more realistic model would be helpful.
> >
> > In summary, it would be good to see this explanation even if it does not result yet in the end-to-end proof, and in my opinion it could show the pathway towards more realistic analysis. If the authors give such convincing links with handling the noise in this real-world setting, I am happy to update my score towards recommending acceptance. I would like to thank again the authors for improving the paper.

---

> > > ### Author Response · Authors · 2021-11-23
> > > **Response to the follow-up question**
> > >
> > > We thank the reviewer for the insightful suggestion. We agree with the reviewer that addressing incorrect labels is a promising direction towards making HPO more robust, and we are currently working on the extension of “label-robust” HPO by leveraging the technique of “robust classification” [Bertsimas et al., 2019]. For example, we could define the loss function of label-robust HPO as
> > > \begin{aligned}
> > >      \hat{L}(\theta)=\max_{I(D_\pi;\hat{A}^{\pi})\leq \alpha \cdot\lvert D_\pi\rvert }\frac{1}{\lvert D_\pi\rvert} \sum_{(s,a)\in D_{\pi}}\lvert \hat{A}^{\pi}(s,a)\rvert \cdot {\max\big\lbrace 0,\epsilon-(\rho_{s,a}(\theta)-1)\text{sign}(\hat{A}^{\pi}(s,a))\big\rbrace}.
> > > \end{aligned}
> > > where $I(D_\pi;\hat{A}^{\pi})$ denotes the number of incorrect labels in the minibatch $D_\pi$ and $\alpha$ denotes the percentage of noisy labels in the worst case.
> > > This formulation naturally provides a class of HPO algorithms that are more robust to the noisy labels due to the estimated advantage function.
> > >
> > > We have highlighted this connection between HPO and robust classification and noisy labels as well as pointed to the key references in Remark 3 of the updated manuscript.
> > > We expect that the theoretical analysis and the proposed algorithms in this paper could help pave the way towards better understanding PPO-clip and policy improvement in even more realistic settings.
> > >
> > > [Bertsimas et al., 2019] Dimitris Bertsimas, Jack Dunn, Colin Pawlowski, and Ying Daisy Zhuo, “Robust Classification,” INFORMS Journal on Optimization, 2019.

---

> > > > ### Comment · Reviewer_vtrj · 2021-11-25
> > > > **Re: Clarification**
> > > >
> > > > To understand this answer better, I have a follow up question. Would the authors also explain, whether and why this formulation with the label noise-resistant loss would be compatible with the existing proofs of Theorem 1? If there is a way to show that by plugging the noise-resistant loss into the HPO model would help lift the Assumption 4, this is a very strong point. If not, it would be good to discuss in this rebuttal response (and then in the next paper edit) what is exactly that won't make this proof possible with noise-resistant loss and Assumption 4.

---

> > > > > ### Author Response · Authors · 2021-11-30
> > > > > **Response to the follow-up question**
> > > > >
> > > > > We thank the reviewer for the follow-up question. Despite that we do not have a formal proof for this label-robust formulation yet, we expect this label-robust formulation to be more accessible than other existing formulations of robust classification since the original HPO loss in Eq(11) is a special case with $\alpha=0$ (where $\alpha$ is the percentage of noisy labels in the worst case). Moreover, the label-robust loss function remains convex since it is the point-wise maximum of multiple convex functions. This allows us to reuse EMDA for finding the minimizer of the loss function, and the nice properties of EMDA can be preserved.
> > > > >
> > > > > To establish the convergence result of the label-robust HPO, one major challenge that we foresee is to show the property of state-wise policy improvement as in Lemma 1, which is mainly based on Lemmas 5-6. Regarding Lemma 6, the technical challenge is to analytically find the minimum of $\hat{L}(\theta)$, which turns out to be non-trivial to express in closed form. Once Lemma 6 is extended for the label-robust formulation, the threshold for EMDA in Line 6 of Algorithm 1 can be determined, and accordingly, Lemma 5 can be established.
> > > > >
> > > > > We will add the above discussion in the next version of the paper.

---

### Official Review · Reviewer_PJFP · 2021-11-02

**Correctness:** 3
**Technical Novelty And Significance:** 2
**Empirical Novelty And Significance:** 2
**Recommendation:** 3
**Confidence:** 5

**Main Review:**


Strengths:
The writing of the paper is clear. The presentations of the theorems are easy to understand.
The interesting part of the paper is to generalize PPO to other classifiers. I suggest the author focus more on it.

Weaknesses:
The major theoretical contribution is the proof of the convergence of PPO. However, in my view, this paper contributes little to understanding PPO. I give my reasons as follows.
- The author proves the result by introducing a very strong assumption (Assumption 3), which in fact includes two assumptions.
	- First, the initial state distribution is strictly positive; thus the policy is possible to sample the state-action pair over the entire space. However, this is almost impossible in practice. This implies that the result of this paper can contribute to a very limited range of tasks.
	- Second, the sampled training dataset are sufficiently large to cover all possible state-action pair.
- The key of PPO is the clipping technique. This is the key to ensuring the performance is bound. In my view, the theoretical result in this paper (Theorem 1) still holds if one removes the clipping function. If I'm right, then this paper contributes little to understanding PPO.
- The author claim that "no additional constraints, such as the KL divergence constraint used in TRPO, are needed to ensure policy improvement". By this claim, do you think that PPO can remove the clipping technique? How do you think the effect of policy constraint in TRPO (KL divergence constraint) and PPO (clipping technique)?

Overall, in my view, the contribution of the paper is to prove the performance improvement of the objective function without clipping technique (by some very strong assumptions).



**Summary Of The Paper:**


This paper reinterprets the theory of PPO-clip based on the hinge policy optimization. They prove the global convergence of PPO by introducing some assumptions. Besides, they generalize the algorithm to a new family of policy-based algorithms by regarding the policy as a generalized classifier.

**Summary Of The Review:**

The paper does a good job in sharing new insights of policy optimization, and in connecting the policy optimization with classification. However, the two main claimed contributions are somewhat weak.
Rethinking policy improvement (Proposition 1 and 2) is not the original idea of this paper. Reinterpreting PPO by hinge loss is also not the idea of this paper. At the same time, the improvement analysis is based on some very strong assumptions. Therefore, I vote for rejection for this paper.

---

> ### Author Response · Authors · 2021-11-21
> **Response to Reviewer PJFP (1/2)**
>
> We thank the reviewer for the constructive feedback for improving our paper. We provide our point-by-point response as follows.
>
> #### (A1) “The key of PPO is the clipping technique. This is the key to ensuring the performance is bound. In my view, the theoretical result in this paper (Theorem 1) still holds if one removes the clipping function. If I'm right, then this paper contributes little to understanding PPO.”
>
> Removing the clipping function of PPO-clip would lead to a loss function as $\sum_{(s,a)} \frac{\pi_{\theta}(a|s)}{\pi(a|s)}A^{\pi}(s,a)$ and result in an algorithm similar to the “one-step greedy policy improvement” in the classic policy iteration.
> It is well-known that such one-step greedy policy improvement can achieve an optimal policy. However, this fact does not imply that PPO-clip can provably attain an optimal policy since adding the clipping function would totally change the convergence behavior.
> As a result, it remains non-trivial to establish the global convergence of PPO-clip.
>
> As also mentioned by Reviewer FV8P, one major contribution of this paper is to provide the first formalization (from the perspective of hinge loss and large-margin classification) which allows a theoretical analysis of PPO-clip.
> Moreover, based on this formalization, we find that the global convergence of HPO requires only the correct signs of the advantage values (rather than the true advantage function required by the policy gradient methods [Agarwal et al., 2019]). This insight could help provide a new perspective for understanding the superior empirical performance of PPO-clip in various domains.
>
> #### (A2) “The author claim that ‘no additional constraints, such as the KL divergence constraint used in TRPO, are needed to ensure policy improvement’. By this claim, do you think that PPO can remove the clipping technique? How do you think the effect of policy constraint in TRPO (KL constraint) and PPO (clipping technique)?”
>
> Thank you for bringing up this profound question. To answer this, we would like to first clarify the difference between the two concepts of policy improvement: “State-wise policy improvement” and “policy improvement in average value.”
>
> Under “policy improvement in average value,” a policy $\pi_{\theta_1}$ is said to improve upon $\pi_{\theta_2}$ if $E_{s\sim \mu}[V^{\pi_{\theta_1}}(s)] \geq E_{s\sim \mu}[V^{\pi_{\theta_2}}(s)]$, where $\mu$ is some distribution over states. This can be viewed as finding improved policies in the “parameter space.” This concept is adopted by the policy gradient methods and various policy optimization methods, including PPO and TRPO.
>
> By contrast, as stated in Definition 1, “state-wise policy improvement” is built on the partial ordering of policies and can be viewed as finding improved policies directly in the “policy space.” This concept has been widely adopted by the classic works on solving MDPs. Moreover, it can be easily verified that “state-wise policy improvement” implies “policy improvement in average value.”
> A related comparison of these two concepts has also been discussed in [Tessler et al., 2019].
>
> As TRPO, PPO-KL, and PPO-clip are designed from the perspective of “policy improvement in average value”, they resort to the KL constraint, KL penalty, and the clipping technique to enforce policy improvement, respectively.
> By contrast, we study HPO (and PPO-clip as a special case of HPO) through the lens of state-wise policy improvement. Propositions 1-2 provide sufficient conditions for “state-wise policy improvement”, and accordingly we leverage this concept to design HPO and prove the convergence of HPO in Theorem 1. Notably, while PPO-clip was originally designed based on  “policy improvement in average value”, we show that the formalization built on the perspective of “state-wise policy improvement” better allows a theoretical analysis of PPO-clip.
>
> [Tessler et al., 2019] Chen Tessler, Guy Tennenholtz, and Shie Mannor, “Distributional Policy Optimization: An Alternative Approach for Continuous Control,” NeurIPS 2019.
>
> #### (A3) Regarding Assumption 3
>
> This paper is meant to take the first step toward establishing the theoretical foundation of the proposed new HPO framework and PPO-clip, and we leverage Assumption 3 in order to rigorously establish the global convergence property. We expect that this could help pave the way toward better understanding PPO-clip in more general settings.
>
> Moreover, as discussed in Remark 3, the recent convergence results of vanilla policy gradient (e.g., [Agarwal et al., 2019]) rely on the true advantage values, which essentially requires visiting all state-action pairs after each policy update. Then, Assumption 3 appears no stronger than the assumption of true advantage values. On the other hand, for problems with a large state space, this assumption could be roughly met by considering a behavior policy with a small chance to act randomly for exploration and performing V-trace for advantage estimation.

---

> ### Author Response · Authors · 2021-11-21
> **Response to Reviewer PJFP (2/2)**
>
> #### (A4) “Overall, in my view, the contribution of the paper is to prove the performance improvement of the objective function without clipping technique (by some very strong assumptions).”
>
> As highlighted in the response to (A1) and (A2), our main contributions are:
> - We rethink policy optimization through the lens of “state-wise policy improvement” (compared to “policy improvement in average value” adopted by TRPO, for example).
> - Accordingly, we connect state-wise policy improvement with large-margin classification with hinge loss (which is equivalent to adding the clipping function) and propose a new class of algorithms called HPO.
> - Built on this, we provide the first formalization (from the perspective of hinge loss and large-margin classification) which allows a theoretical analysis of PPO-clip (as a special case of HPO).

---

> ### Comment · Reviewer_PJFP · 2021-11-27
> **After reading the rebuttal**
>
> Thanks for your response and your understanding.
>
> I agree and believe that the author did prove the convergence of loss objective with clip (i.e., PPO-CLIP). And I appreciate that the paper did provide some interesting insights, e.g., not requiring exact advantage value, extending to any classifiers.
> However, some of my main concerns are not well addressed in the authors' responses.
>
>
> ### the insight in understanding PPO
> The authors claim that
> > This insight could help provide a new perspective for understanding the superior empirical performance of PPO-clip in various domains."
>
> which is also the main claim in their abstract and introduction. However, my point is that, does using clipping bring any further benefits on the convergence results? To be specific, I list questions below.
>
> 1. Does the objective without clip can also guarantee convergence? Do the results still holds without a clip in this paper? (It would be appreciated if the author could list the theorems which do not hold without the clipping.)
> 2. If the answer to question 1 is yes, what is the necessity to use a clip? Why is the clipping useful? Does using a clip bring any benefits to your results?
> 3. Does the result in this paper provide any insight into the necessity of using a clip? If the answer is not, then in my view it's exaggerated to claim that this paper provides a new perspective for understanding the superior empirical performance of PPO-clip.
>
> I am happy to increase my score and would appreciate it if the author could address these concerns.
>
>
> ### Discussion about  TRPO and PPO
> Thanks for sharing your understanding of the theoretical results of TRPO and PPO.
> One thing that the author did not mention is that TRPO and PPO-penalty provide a lower performance bound $\eta(\pi_{new}) \geq L(\pi_{new}) \geq \eta(\pi_{old})$. If one can improve $\eta(\pi_{old})$ by a large margin, then a large-margin improvement by the author can be obtained. This is a more strong result compared to just improvement ($\eta(\pi_{new}) \geq \eta(\pi_{old})$).  Can the author also provide such a result? I do not expect the author to come up with proof in rebuttal. It would be appreciated if the author could share your understanding of this problem.
>
>
>
>
> ### Minor Discussion
> As stated by other reviewers, assumption 4 is strong. I believe that it is hard to remove this assumption, as it is hard to guarantee Proposition 1 (eq. 5) under label-noise. However, this is the most interesting part to me that one only requires a correct sign rather than an accurate value. By the way, the PPO uses GAE as the advantage value, which is very noisy and inaccurate. I think the paper did provide insight into this setting.

---

> > ### Author Response · Authors · 2021-11-30
> > **Response to the follow-up discussion (1/2)**
> >
> > ### **1. Regarding the benefits of the clipping function**
> > The clipping function in PPO-clip was originally proposed to enforce strict policy improvement in terms of average value [Schulman et al., 2017], with an aim to avoid policy degradation during training due to an improper step size under the policy gradient methods.
> > One key insight provided by our paper is that the clipping function enables HPO (and PPO-clip as a special case) to be “*robust to inaccurate advantage estimation*” in the sense of achieving state-wise policy improvement (and thereafter attaining global convergence) given that the advantage signs are correctly recovered.
> > By contrast, even in the tabular case, the property of state-wise policy improvement is not guaranteed to hold under the standard (projected) gradient update without the clipping function.
> >
> > E.g., one can consider a simple one-state MDP (say 1 non-terminal state $s$ and 1 terminal state $s’$) with three actions (denoted by $a_1, a_2, a_3$) under tabular policy parameterization:
> > - Suppose the transition probabilities are $P(s’\rvert s,a_i)=1$, for $i=1,2,3$, and the reward function is $r(s,a_1)=1$, $r(s,a_2)=-0.5$, and $r(s,a_3)=-3$.
> > - For ease of notation, let $\pi$ denote the current policy and $\pi_i$ denote the probability of action $a_i$ at state $s$. Suppose the current parameters are $\pi_1=0.5$, $\pi_2=0.4$, $\pi_3=0.1$. Then, we have $V^{\pi}(s)=0$, $A^{\pi}(s,a_1)=1$, $A^{\pi}(s,a_2)=-0.5$, and $A^{\pi}(s,a_3)=-3$.
> > - Under the standard (projected) gradient update without clipping, the objective is $J(\theta)=\sum_{i=1}^{3} \frac{\pi_\theta(a_i\rvert s)}{\pi(a_i \rvert s)} \hat{A}(s,a_i)$, where $\hat{A}(s,a_i)$ is the estimated advantage.
> > - Let $\theta_{i}'$ and $\theta_{i}''$ denote the updated policy parameters before and after the projection onto a simplex, respectively. That is, $\theta_{i}'=\pi_i +\eta \frac{\partial J} {\partial \theta_i}$ (where $\eta$ is the step size), and $\theta_{i}^{''}$s are the parameters obtained by projecting $\theta_{i}'$s onto a simplex. Set $\eta=0.01$.
> >
> > |$i$| $\pi_i$|$A^{\pi}(s,a_i)$|$\hat{A}(s,a_i)$| $\theta'_i$| $\theta^{''}_i$|
> > |-|---|---|---|---|---|
> > |1 | 0.5 | 1| 0.1 | 0.5020 | 0.5463 |
> > |2 | 0.4 | -0.5 | -5 | 0.2750 | 0.3193 |
> > |3 | 0.1 | -3 | -0.1 | 0.0900 | 0.1343 |
> >
> > As a result, $V^{\pi_{\theta''}}(s)$ of the updated policy is -0.0163 < 0.
> > Based on the above, we can see that policy improvement is not guaranteed under the standard projected gradient update without clipping.
> >
> > By contrast, Lemma 1 shows that state-wise policy improvement can be achieved under HPO given the correct advantage signs. This shows that the clipping (interpreted as hinge loss in our paper) provides a systematic way to enforce policy improvement under inaccurate advantage.
> >
> > ### **2. Regarding whether the analysis of HPO-AM holds without clipping**
> > Note that for HPO with a constant margin $\epsilon$, an infinitely large $\epsilon$ would be equivalent to the case of "no clipping". Below we go ahead and discuss HPO with a constant margin:
> >
> > Recall from Eq(13) that in HPO-AM the margin of hinge loss is adaptive based on the probabilities of the actions with positive and negative advantage values.
> > We did a check on the analysis, and it appears feasible to extend the proof of HPO-AM to constant margins, i.e., $\epsilon^{(t)}\equiv \epsilon$:
> > (Note that all the reference numbers of equations and lemmas below are with respect to the updated manuscript)
> > - If the margin is a constant, the original Lemmas 5-6 would no longer hold since the policies selected by Eq(24) and Eq(27) may not exist (i.e., $(1+\zeta \epsilon)\pi(a\rvert s)$ in Eq(24) and $(1+\alpha \epsilon)\pi(a\rvert s)$ in Eq(27) can be greater than 1). As a result, to establish Lemmas 5-6 for constant margin, one would need to first revise Line 6 in Alg. 1 and replace the EMDA threshold (i.e., $(1-\zeta)\delta^{(t)}\epsilon^{(t)}$) with a proper one that depends on the minimum of $\hat{L}(\theta)$ as well as $\epsilon$. Accordingly, the state-wise policy improvement in Lemma 1 and the property of Lemma 2 could still hold.
> > - Under a constant margin, Lemma 3 shall hold directly.
> > - Lemma 4: Eqs(50)-(62) still hold under a constant margin. Hence, the result in Eq(65) shall also hold.
> >
> > One drawback of using a constant margin (and “no clipping” as a special case) is that the algorithm could easily commit to deterministic policies with sub-optimal actions. E.g., for the case of no clipping, to maximize $\sum_{(s,a)}\frac{\pi_\theta(a|s)}{\pi(a|s)}\hat{A}(s,a)$, one would assign probability 1 to the action with the largest $\frac{\hat{A}(s,a)}{\pi(a|s)}$ (whereas this action can be sub-optimal). Similar arguments would apply for finite constant margins.
> > Such early committing behavior can greatly affect the accuracy of estimated advantage due to insufficient exploration (given that PPO-clip and HPO are on-policy). Hence, it is algorithmically more appropriate to use an adaptive margin for HPO.

---

> > ### Author Response · Authors · 2021-11-30
> > **Response to the follow-up discussion (2/2)**
> >
> > ### **3. Could the margin of policy improvement (i.e., $\eta(\pi_{new})-\eta(\pi_{old})$) be quantified under HPO?**
> >
> > Yes, the margin of policy improvement can be derived from the analysis of Lemma 5 in the updated manuscript. Specifically, Eqs(24)-(25) explicitly characterize the bounds on the amount of change in the action probabilities, and accordingly one could obtain the margin of policy improvement by applying the performance difference lemma in Eq(4). The resulting lower bound on the margin of policy improvement of HPO would depend on the margin $\epsilon^{(t)}$, just like TRPO and the size of the corresponding KL neighborhood.

---

> ### Comment · Reviewer_PJFP · 2021-11-27
> **Further explanation**
>
> My key point is that, if my conjectures are correct (the theoretical results in the paper also hold for the objective without clipping), then the paper provides the same theoretical results of PPO-clip as those of the objective without clipping. This implies that it provides no new insights to understand why PPO-clip is useful.

---

### Decision · Program_Chairs · 2022-01-20

**Decision:**

Reject

**Comment:**

All reviewers agreed that analysis of PPO is interesting.
During the discussion, however, there was an agreement that the current work is too thin in novelty and contribution: it provides only convergence analysis under very strong assumptions, and heavily builds on techniques from prior works. Meanwhile, for conventional policy gradient, recent works provided convergence rates.
As one reviewer pointed out - this work does not further our theoretical understanding on why PPO is better than vanilla policy gradient, as all the established results hold for policy gradient, even with less assumptions.
I encourage the authors to strengthen their paper by relaxing Assumption 4 (perhaps based on the robust classification idea raised in the discussion), and by further providing rate results.